# Whole-genome sequencing of 1,171 elderly admixed individuals from Brazil

Michel S. Naslavsky [1,2,3,29 ✉], Marilia O. Scliar[1,29], Guilherme L. Yamamoto[1,4,5,6,29], Jaqueline Yu Ting Wang[1], Stepanka Zverinova [7], Tatiana Karp[7], Kelly Nunes[2], José Ricardo Magliocco Ceroni[1], Diego Lima de Carvalho [1], Carlos Eduardo da Silva Simões[1], Daniel Bozoklian[1], Ricardo Nonaka[1], Nayane dos Santos Brito Silva[8], Andreia da Silva Souza[8], Heloísa de Souza Andrade [8], Marília Rodrigues Silva Passos[8], Camila Ferreira Bannwart Castro[8,9], Celso T. Mendes-Junior [10], Rafael L. V. Mercuri[11,12,13], Thiago L. A. Miller[11,12], Jose Leonel Buzzo[11,12], Fernanda O. Rego[11], Nathalia M. Araújo[14], Wagner C. S. Magalhães[14,15], Regina Célia Mingroni-Netto[1,2], Victor Borda[14], Heinner Guio[16,17], Carlos P. Rojas[16], Cesar Sanchez[16], Omar Caceres [16], Michael Dean [18], Mauricio L. Barreto[19,20], Maria Fernanda Lima-Costa[21,22], Bernardo L. Horta [23], Eduardo Tarazona-Santos[14,24,25,26], Diogo Meyer[2], Pedro A. F. Galante [11], Victor Guryev [7], Erick C. Castelli[8,9], Yeda A. O. Duarte[27,28], Maria Rita Passos-Bueno[1,2] & Mayana Zatz[1,2 ✉]

As whole-genome sequencing (WGS) becomes the gold standard tool for studying population genomics and medical applications, data on diverse non-European and admixed individuals are still scarce. Here, we present a high-coverage WGS dataset of 1,171 highly admixed elderly Brazilians from a census-based cohort, providing over 76 million variants, of which ~2 million are absent from large public databases. WGS enables identification of ~2,000 previously undescribed mobile element insertions without previous description, nearly 5 Mb of genomic segments absent from the human genome reference, and over 140 alleles from HLA genes absent from public resources. We reclassify and curate pathogenicity assertions for nearly four hundred variants in genes associated with dominantly-inherited Mendelian disorders and calculate the incidence for selected recessive disorders, demonstrating the clinical usefulness of the present study. Finally, we observe that whole-genome and HLA imputation could be significantly improved compared to available datasets since rare variation represents the largest proportion of input from WGS. These results demonstrate that even smaller sample sizes of underrepresented populations bring relevant data for genomic studies, especially when exploring analyses allowed only by WGS.

A full list of author affiliations appears at the end of the paper.

Whole-genome sequencing (WGS) of a large number of individuals can reveal rare variants in known disease genes[1–4], improve identification of novel genes and pathways associated with phenotypes[5] and identify genomic regions not represented on reference genomes[6]. Ancestry diversity is critical to elucidate differences in disease's genomic architecture and improve signals detected by previous studies since non-European and admixed populations harbor-specific variants[7–9], which are still vastly underrepresented in genomic studies[10]. The lack of diversity leads to a significant bias on the primary resource for precision medicine and consequently less accurate tests on non-European descent individuals, potentially increasing health disparities[10–12].

Knowledge about allelic frequencies from multiple populations is also crucial when prioritizing candidate clinical variants. For rare Mendelian disorders, the frequency of a pathogenic variant in any given population cannot be higher than the incidence of its associated disease, considering compatibility with a mode of inheritance and penetrance[13–15]. Moreover, the penetrance of variants may vary across backgrounds[16,17]. For variants associated with monogenic early and adult-onset disorders, unaffected elderly individuals serve as a proper control group to improve diagnosis accuracy. Since many diseases manifest later in life, datasets composed by adults can include carriers that may express some or full clinical phenotypes. Even studies on late-onset diseases can be powered by a control group of verified unaffected status when aged older than the average age at onset. This rationale was previously explored by us using whole-exome sequencing of elderly Brazilians[18], and by others using a European-descent whole-genome dataset of Australian elderly[19].

Here we present a high-coverage WGS of a Latin American census-based cohort composed of 1171 unrelated elderly from São Paulo, Brazil's largest metropolis. Among the residents, there are immigrant descendants from different continents, individuals from various Brazilian states[20,21], and 66 individuals born abroad, mostly in Europe and Japan. These individuals aged 60 or older have been comprehensively phenotyped by the longitudinal Health, Well-Being, and Aging (SABE - *Saúde, Bem-estar e Envelhecimento*) study[21]. By carrying out WGS on this population-based cohort, we identified genomic variation absent from public databases, including single nucleotide substitutions, insertion/deletion variants (indels), chromosomal haplotypes, accurate HLA variant calls, mobile element insertions, and non-reference sequences (NRS). Additionally, we explored pathogenicity assertions in disease-related genes of clinical relevance. We also created reference imputation panels for the whole-genome and HLA alleles, which improved imputation accuracy. Lastly, we provide variants and respective allelic frequencies in a public resource, ABraOM (http://abraom.ib.usp.br).

## Results

**Cohort description**. SABE is a longitudinal study initiated in 2000, with a follow-up occurring every 5 years (see Supplementary

Information and Supplementary Fig. 1 for details on study design). After quality control, 1171 unrelated individuals composed the WGS dataset, with an average age of 71.86 (±7.94) years and 1.74 female to male ratio (Supplementary Table 1). Data collection[21] involves at-home interviews with 11-section questionnaires, including cognitive screening, self-reported race/ethnicity status and standard tests of over 20 health conditions, habits and phenotypes, medication inventory, and functional measurements, such as frailty, dexterity, balance, and mobility summarized in Supplementary Table 2.

High-coverage WGS data (average $38.6 \pm 6\times$) was generated using a previously described protocol[4] and analyzed (Supplementary Fig. 2, Supplementary Data 1). Nearly 76 million single nucleotide variants (SNVs) and indels were identified with their predicted consequences, including over 22 thousand potential loss of function (pLOF) variants annotated by LOFTEE[3] (Supplementary Table 3). After filtering out low confidence variants (Methods, Supplementary Fig. 3), we obtained a dataset of over 61 million variants, among which ~2 million are not described in gnomAD, dbSNP, or 1000 Genomes (Supplementary Fig. 4).

The average global ancestries for SABE are $0.726 \pm 0.263$ European, $0.178 \pm 0.209$ African, $0.067 \pm 0.066$ Native American, and $0.028 \pm 0.162$ East Asian (Fig. 1, Methods, Supplementary Table 4). There is considerable variation in individual ancestries, ranging from a single ancestry to admixture involving two or more ancestries (~75% of the cohort). Individuals with East Asian ancestry have virtually 100% of this parental component, consistent with the historical information as the first generation of Japanese immigrants (Fig. 1 and Supplementary Fig. 5). As part of the interview process, all individuals were asked to self-report to one of the ethnoracial groups routinely used by the Brazilian Institute of Geography and Statistics in the national census[22] (White, Black, Yellow, *Pardo*—translated as Mixed, or Indigenous). The average proportions of genetic ancestry significantly vary among self-reported ethnoracial groups (one way ANOVA *p*-value < 0.0001; Supplementary Fig. 5A). Yet, 37% of the variation in European ancestry among individuals is not explained by self-reported ethnoracial groups ($r^2 = 0.63$; *p*-value < 2.2e-16; fit linear model d.f. 1083). Thus, although there is a correlation between genetic ancestry and self-reported ethnoracial groups, they are not able to capture accurate information about the heterogeneity and proportions of individuals' genetic ancestry. In addition, three individuals self-reported as Indigenous had a high degree of admixture but were removed from Fig. 1 due to the small sample size of the group.

**Clinically relevant findings**. Although SABE participants are not affected by severe monogenic disorders, they might carry pathogenic variants related to recessively inherited disorders, mild phenotypes, or incomplete penetrance. Moreover, it is known that many pathogenic assertions are misclassified[15], and

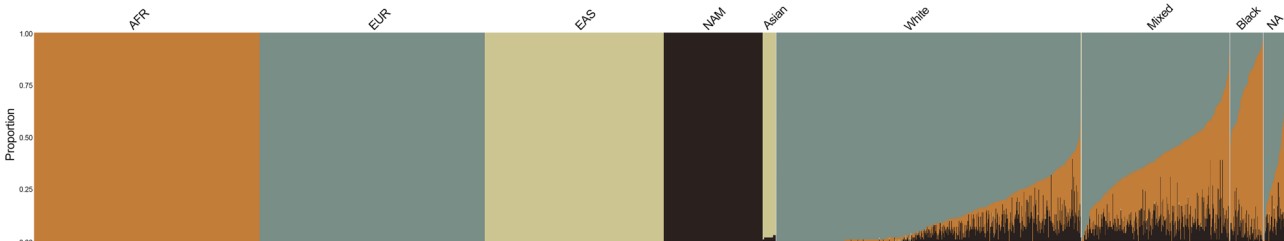

**Fig. 1 Global ancestry inference of SABE cohort.** Individual ancestry bar plots of SABE cohort (N = 1168) using supervised admixture analysis (K = 4). Africans (AFR), Europeans (EUR), East Asians (EAS), and Native Americans (NAM) samples are used as parental populations. SABE cohort individuals are distributed by self-reported ethnoracial groups (according to the Brazilian Institute of Geography and Statistics categories[22] Asian, White, Mixed, and Black; see Supplementary Fig. 5). NA not available.

cohorts with individual genotypes and phenotypes can aid reclassification.

We analyzed 'Pathogenic' or 'Likely Pathogenic' (P/LP) ClinVar asserted variants carried by SABE individuals across 4250 genes associated with monogenic disorders (Online Mendelian Inheritance in Man—OMIM disease genes, Supplementary Data 2) and manually curated the variants in genes associated with dominant inheritance using ACMG guidelines[13] and in-depth literature support, performed by clinical geneticists (complete workflow on filtering, annotation and counts are described in Supplementary Information Note 4, Supplementary Figs. 6–9, Supplementary Tables 5–8 and Supplementary Data 3–6. In total, out of 394 variants asserted as either P/LP in genes annotated to have at least one phenotype with dominant inheritance, curation resulted in the reclassification due to pathogenicity downgrade (116 or 29% of variants; Supplementary Table 8, Supplementary Data 4), or compatible categories with unaffected phenotypes due to inheritance mechanism (53%) or incomplete penetrance (14%), with only 3% of variants associated with a matching detectable phenotype (Supplementary Table 8, Supplementary Data 7).

It has been reported that large datasets contain pathogenic mutations that can be harbored by unaffected individuals, as shown by Chen and colleagues after deeply screening genes associated with monogenic early-onset disorders[23]. Healthy elderly individuals from Australia are reportedly depleted of disease-causing variants but still carry clinically relevant mutations[19]. It is noteworthy that pathogenicity misclassification itself can alter estimations of disease prevalence[15]. Manual curation promotes the downgrading of P/LP assertions when larger sample sizes and diverse ancestries are added to databases, which may increase the frequency of some variants, allowing updates of older assertions that are likely to have an inflated proportion of P/LP assertions[15,24]. Reclassification of variants is improved when based on standardized criteria and reports of reduced penetrance[14,15]. Moreover, variants' penetrance may differ according to different genetic or environmental backgrounds[17], observable in well-established monogenic mutations that segregate in families[25] that can be modified by rare variants[26] or a polygenic profile[27]. This explains why population-specific genomic architecture reduces GWAS replication[28] and affects the distribution of polygenic risk scores[9,29]. Therefore, pathogenicity assertions must be interpreted based on specific population datasets. Also, regarding P/LP asserted variants in the 59 ACMG actionable genes list[30] (Supplementary Data 6), 14 were found in 1.2% of individuals compared to the 1.1% found in the Australian elderly cohort[19], demonstrating that clinically relevant variants are detectable at low but equivalent proportions.

Common pathogenic variants in genes associated with selected recessively inherited Mendelian disorders were manually curated using locus-specific databases and ACMG. Common and rare P/LP variants in *CFTR*, *HBB*, *GJB2*, *MEFV*, and *HFE* were accounted for incidence estimates after calculating, from carrier frequencies, the expected offspring number of homozygotes and compound heterozygotes (Supplementary Data 7). We showed that cystic fibrosis and hemoglobinopathies have similar expected incidences when compared to gnomAD, respectively, about one cystic fibrosis affected newborn in ten thousand births and one hemoglobinopathy affected newborn in three thousand. Estimations were calculated from *CFTR* pathogenic variant carrier frequencies of 1.8% in SABE and 2% in gnomAD (Chi-squared 0.26, 1 d.f. = 1, $p = 0.63$) and *HBB* of 3.9% in SABE and 3.4% in gnomAD (Chi-squared 0.26, 1 d.f. = 1, $p = 0.35$). Other diseases appear more frequently in Brazilians (*GJB2*-related deafness, one in 5.7 thousand in SABE versus one in 19 thousand in gnomAD and *MEFV* Familial Mediterranean fever, one in 55 thousand

versus one in 353 thousand in gnomAD). These disparities observed for *GJB2* and *MEFV* between Brazilians and global gnomAD, but similar to gnomAD Latinos (one in 66 thousand for *MEFV*) and PAGE Study samples of Cubans, Puerto Ricans, and Central Americans are probably due to the Iberian, Mediterranean, and Middle Eastern contributions[31–33] in Brazil, but we cannot exclude that penetrance of such variants may be lower than previously estimated.

Estimating the incidence of recessive disorders is challenging due to the ascertainment of unrelated individuals within a given population-based sample and classification of pathogenicity, since most pathogenic variants are rare and the distribution of variants within a population is not known a priori[34]. Our results are limited to known alleles curated by locus-specific databases to provide a comparison of expected individuals in homozygous and biallelic states for selected recessive disorders.

Finally, regarding the potential loss of function variants (pLOFs) within the OMIM disease genes, we identified 3704 non-benign variants (Supplementary Fig. 7), most absent from ClinVar with frequencies comparable to gnomAD. The few but very discrepant frequencies are mostly false positives due to calling or annotation from either dataset (Supplementary Figs. 8, 9).

**Mobile elements insertions (Meis).** We investigated structural variations caused by insertions of mobile elements (MEIs), which constitute a rich and underexplored source of genetic variation. MEIs here identified are insertions to the human reference genome (GRCh38) occurring in at least one out of 1171 SABE genomes. First, we found a set of 7490 nonredundant MEIs, including 5971, 1131, 375, and 13 events of *Alu*, L1, SVA (SINE-R, VNTR, and *Alu* composite), and Human Endogenous Retrovirus K (HERV), respectively (Fig. 2A, variants deposited in http://abraom.ib.usp.br). As expected[35], *Alu*, and L1 insertions are the prevalent events (94.7%). Next, we classified these MEIs into (i) Shared (i.e., MEIs present in two or more unrelated SABE individuals and also in individuals from gnomAD); (ii) SABE-private events (present in two or more SABE genomes, but absent in other genomes from Database of Genomic Variation – DGV[36], which include gnomAD data); and (iii) Singletons (present in only one SABE individual and absent from DGV. Shared is the most frequent class, corresponding to 5571 (74.3%) MEIs (Fig. 2B). SABE-private MEIs constitute 1501 (20.1%) events (Fig. 2B) and comprise ~0.97 kbp potentially polymorphic and still unreported events in other databases. We also found 418 insertions classified as Singletons (5.6%; Fig. 2B), which are either somatic or germinative MEIs. On average, each individual carries 869 MEIs (Fig. 2C), among which the vast majority (97.0%) are *Alu* (758 events, on average) and L1 (85 events, on average), Fig. 2C. As expected, most MEIs per individual are Shared (774 (89.0%); Fig. 2C). Furthermore, individuals from our cohort carry 10.9% of events classified as SABE-private (Fig. 2C), which presented a lower allele frequency in comparison to the class of common events (Fig. 2D; *p*-value = 1.4e-0.7; Mann-Whitney test). Even though we expected Shared MEIs to have a higher allele frequency, 103 (7.3%) of SABE-private events presented an unexpected high allele frequency (>20%). Further validations are required to confirm if these MEIs are enriched events in our cohort (and absent in other populations) or calling artifacts.

Next, we examined the insertion profile of MEIs regarding their genomic locations. We observed: (i) a positive correlation between the number of MEIs and the chromosome length (Supplementary Fig. 10); (ii) that L1 and *Alu* insertions are skewed to AT-rich regions, while HERVs are biased to GC-rich regions (Supplementary Fig. 11); (iii) an enrichment of MEIs into

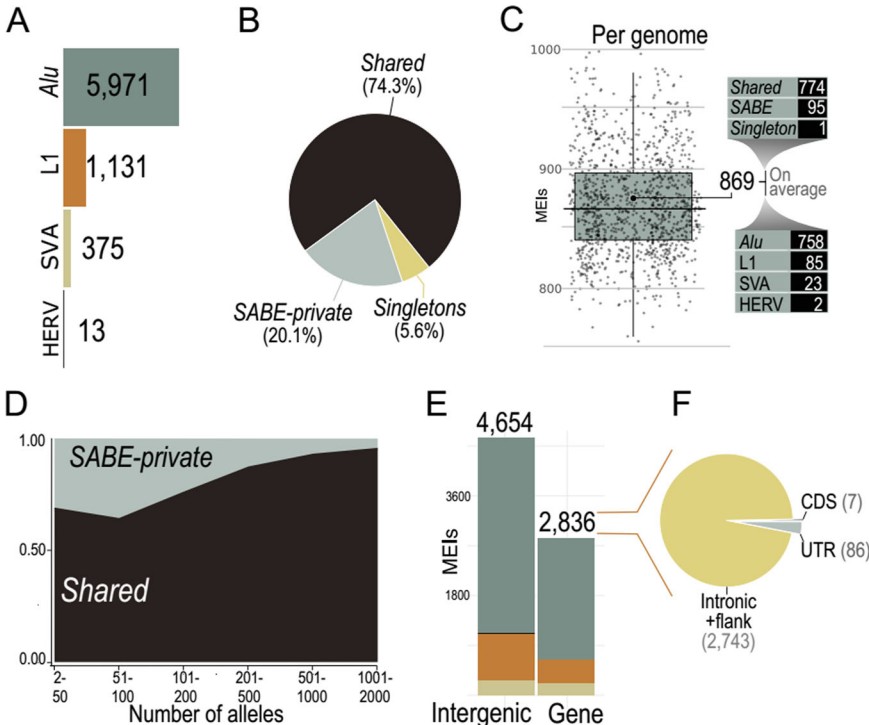

**Fig. 2 A landscape of mobile element insertions (MEIs) into SABE genomes. A** Total of MEIs in SABE genomes. As expected, Alu and L1 elements are predominant elements. **B** Proportion MEIs in Shared (present in DGV genomes), in two or more genomes from SABE cohort (SABE-private) and present in only one SABE genome (Singletons) **C** Number of MEIs per individual. The lower and upper hinges correspond to the 25th and 75th percentiles, respectively, and the whiskers represent the 1.58 × interquartile range (IQR) extending from the hinges. **D** Distribution of allele frequencies of Shared and SABE-private MEIs. **E** Number of MEIs into genes and in intergenic regions. **F** Number of MEIs in the coding region (CDS), untranslated regions (UTR), or intronic and flank (2 kbp near genes).

intergenic regions (Fig. 2F; two-sided $p$-value < 0.00001; chi-square = 72.608; d.f. = 1). Out of the 2836 MEIs within genic regions, intronic regions have significantly more (2743) MEIs than untranslated (UTRs: 86) and protein-coding (CDS: 7) regions ($p$-value < 0.00001; chi-square = 62.3; d.f. = 1), suggesting selection against insertions in coding (CDS) or regulatory (UTR) regions. Even though 26 events fall within exonic regions of clinically relevant genes (Supplementary Table 9; Supplementary Data 8), none are described to have phenotypic manifestations in one copy or with loss of function mechanism (Supplementary Information).

**Non-reference sequences (Nrs).** WGS data from diverse human populations can contribute with genomic insertions that are not part of the current reference genome, so-called non-reference segments[6,37]. These mostly uncharacterized sequences contain gene exons and full genes, and may modulate susceptibility and prevalence of different diseases. We characterized these 'missing' segments by performing de novo assembly of high-quality reads that do not map to current reference using a pipeline of assemblers, aligners and mappers, with parameters set to reduce false positives (Methods, Supplementary Fig. 12).

The total lengths of NRS per individual ranged between 11.3 and 23.4 Mbps, with an average of 15.4 Mbps (Supplementary Fig. 13A). The nonredundant non-reference segments library of the SABE dataset contains 192,183 sequences (67.4Mbps), from which 428 NRS (0.43Mbps) were observed in all individuals (Supplementary Fig. 13B). Although most NRS (92.5%, totaling 56.4 Mbps) are shorter than 500 bps, we observed 40 contigs larger than 10 kbps, up to a maximum length of 34.5 Kbps (Supplementary Fig. 13C).

Comparison with NRS from the Chinese HAN population[37], African pan-genome[6], Genome of the Netherlands[38], and NCBI nonredundant database revealed a sizable fraction of 28,264 NR-contigs (totaling 9 Mbps) is unique to the SABE dataset. Simultaneously, as much as 15 Mbps of NR-contigs are shared with the HAN and African Pangenome data (Fig. 3A).

In total, we were able to localize 78,831 contigs (28.2 Mbps) to the most recent reference assembly GRCh38, from which 12,617 localized contigs (4.9 Mbp) are unique to our dataset (Fig. 3B). We have mapped three randomly selected samples and compared results of mapping against GRCh38 and GRCh38 appended with SABE non-reference segments (GRCh38 + SABE_NRS). Expectedly, aligning against extended reference decreased the number of unmapped reads by 2.6% (range from 2.4 to 2.8%). Further, the primary alignments of the reads against GRCh38 + SABE_NRS showed an increased proportion of mapped bases, by 0.3% (0.2–0.3%) and decreased number of soft-masked (8.4%, 7.0–9.8%), deleted (9.9%, 8.4–11.0%), and inserted bases (34.4%, 29.1–39.6%), indicating that an extended reference reduces the number of misalignments. Additional investigation is required, including calling variants in Brazilian samples using the GRCh38 reference appended with SABE.

The reported population frequency and genomic location of these non-reference segments will assist future functional studies that characterize their contribution to protein isoforms, gene regulation, and their potential link to human diseases[39].

**An improved Latin American imputation panel.** Previous studies have shown that using a reference panel composed of individuals with a similar genetic background to the target sample improves imputation accuracy, especially for rare variants[40]. We created an imputation panel by merging SABE and the public

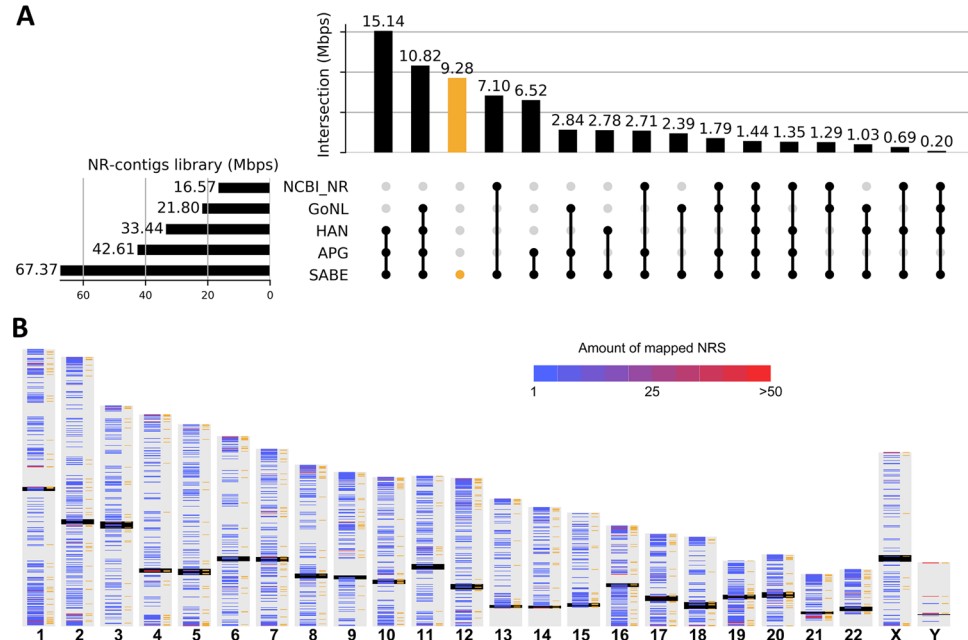

**Fig. 3 Non-reference genome sequences (NRS) in the SABE dataset. A** UpSet plot showing the presence of the SABE NRS in other public databases (sharing among datasets indicated by connected dots): NCBI nonredundant database (NCBI_NR), Genome of the Netherlands (GoNL), NAH Chinese (HAN), and African (APG) pan-genomes. **B** Distribution of NRS across chromosomes. The black bars mark centromeres, bands on the left of each chromosome show density of NRS contigs, orange bands on the right side of each chromosome indicate positions of SABE-private NRS. Chromosome representations are not in scale.

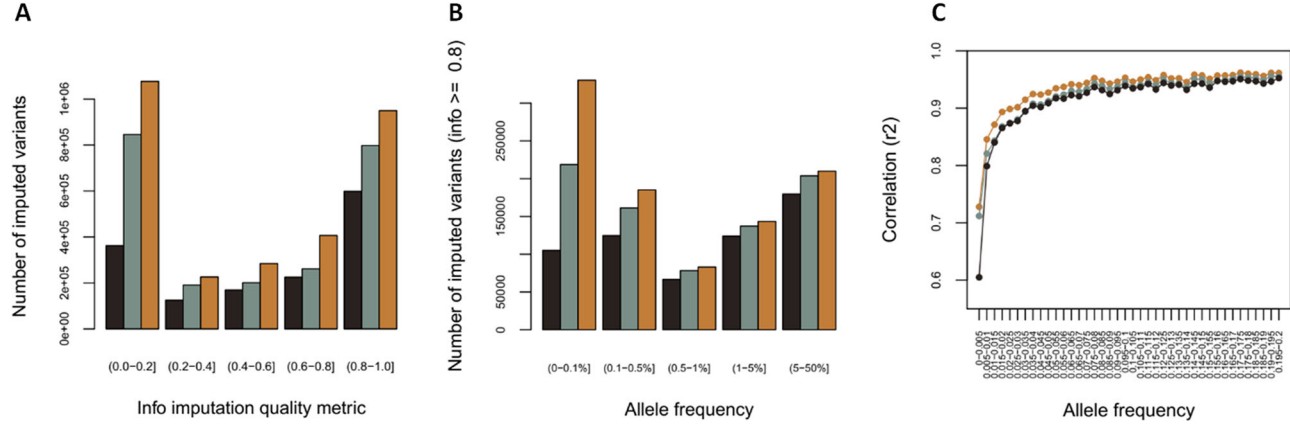

**Fig. 4 Comparison of imputation performance of SABE, 1KGP3, and SABE + 1KGP3 reference panels using the Omni 2.5 M array data for 6487 Brazilians from EPIGEN as target panel (chromosome 15). A** The total number of imputed variants across different classes of info score quality metric. **B** The total number of imputed variants with info score ≥0.8 across the allele frequency spectrum. **C** Improvement in imputation accuracy as a function of minor allele frequency (MAF) for the target dataset after imputation (MAF from 0 to 0.2, bin sizes of 0.005). Similar results were reached for the other chromosomes tested and for each cohort (Supplementary Figs. 14–36; Supplementary Tables 10-16).

1000 Genomes Project Phase 3 dataset (1KGP3)[41], hereafter called the SABE + 1KGP3 reference panel (Supplementary Table 10). Data from chromosomes 15, 17, 20, and 22 were used to test the usefulness of the SABE + 1KGP3 reference panel compared to the 1KGP3 alone. We imputed a dataset of Omni 2.5 M Illumina array genotyped on 6487 Brazilians from the EPIGEN initiative, which is composed of three different cohorts across the country (Salvador, Bambuí, and Pelotas), that vary in admixture levels and demographic histories[42]. When using the SABE + 1KGP3 reference panel, we imputed the largest number of variants, ~20% of which were added exclusively by the SABE dataset (Fig. 4A). There was a gain of ~8% of high-confidence imputed variants (info score > 0.8) by the SABE + 1KGP3

reference panel compared to 1KGP3 alone (Fig. 4B), driven mainly by very rare variants (Fig. 4B), which also mainly contributed in improving imputation accuracy measured by $r^2$ increase (Fig. 4C). We also evaluated the improvement of the SABE + 1KGP3 panel independently in each EPIGEN cohort and in two other admixed Latin American populations from Peru ($N = 391$ Mestizos[43]) and Guatemala ($N = 640$ individuals, unpublished dataset) and also observed a general improvement of imputation (Supplementary Tables 11–16, Supplementary Figs. 14–36), although reduced for Peruvians and Guatemalans when compared to the gain observed for Brazilian EPIGEN cohorts. This improvement was also observed regardless of the chromosome tested.

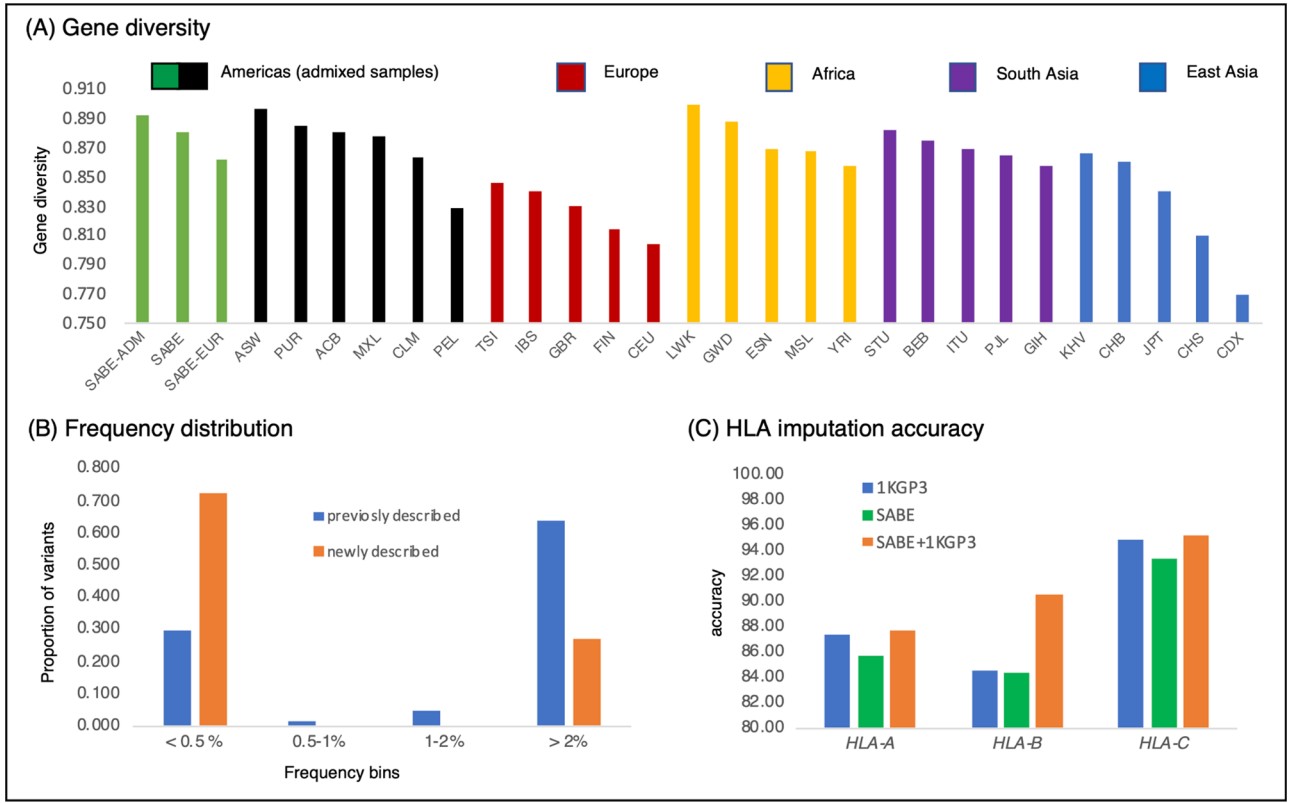

**Fig. 5 HLA polymorphism in the SABE cohort.** SABE and 1KGP3 samples were processed with the same HLA workflow, as described in the Supplementary Information. **A** Average gene diversity across SABE and the 1KGP3 populations considering haplotypes of all SNVs, i.e., the 2064 SNVs from six HLA class I genes, *HLA-A, HLA-B, HLA-C, HLA-E, HLA-F*, and *HLA-G*. SABE all samples from SABA dataset, SABE-ADM samples with at least 30% of both European and African global ancestry, SABE-EUR samples with 100% European global ancestry. **B** The proportion of previously and newly described SABE HLA SNVs according to different minor allele frequency classes. **C** HLA imputation accuracy when using the 1KGP3 (blue), SABE (green), and combining both (orange). Imputation was performed on 146 highly admixed Brazilians previously genotyped on Axiom Human Origins array and HLA genotyping by sequence-based typing methods.

**Diversity of HLA genes**. We previously developed hla-mapper[44] to optimize mappings for HLA genes, providing high-confidence genotype and haplotype calls for this unusually polymorphic region[45], with complex structure involving duplications. We applied hla-mapper in the SABE dataset, detecting 2.4× more variants in the HLA class I genes than with the computational workflow for genotype calling used in the entire genome. We identified an abundance of rare variants not previously described (Fig. 5B) and haplotypes (Supplementary Fig. 37), defining 143 HLA alleles without previous descriptions, mostly rare.

While only 1% of the SABE individuals carry sequences that code for previously undescribed HLA proteins for at least one HLA class I locus, 33% have at least one new sequence comprising introns, exons, and UTRs. Moreover, 2.9% of variants detected in the HLA class I loci are described here and absent in dbSNP, concentrated in introns and regulatory sequences. The list of HLA variants and their frequencies are available in the ABraOM database (http://abraom.ib.usp.br).

To contextualize our findings, we compared polymorphism for the full sequence of *HLA-A, HLA-B, HLA-C, HLA-E, HLA-F*, and *HLA-G* in the SABE to 26 populations from the 1000 Genomes Project (1KG3P), processed through the hla-mapper pipeline. A highly admixed subset of SABE individuals (with at least 30% of both European and African global ancestry, $n = 207$, SABE-ADM, Fig. 5A) presented the third-highest worldwide gene diversity, and second-highest allele richness and mean number of observed haplotypes. The subset of individuals with 100% European global ancestry ($n = 152$, SABE-EUR, Fig. 5A) had

the lowest diversity among subsets we explored within SABE, although still higher than that of individual European populations from 1KGP3. These results highlight not only the contribution of non-European admixture to HLA polymorphism in Brazilians, but also the presence of European ancestries (such as Iberic and Mediterranean) that are likely to be underrepresented in major databases.

Finally, we used SABE as part of a reference panel to impute HLA alleles in a sample of 146 highly admixed Brazilians from another study[46]. As for the whole-genome imputation, the SABE + 1KG3P combined reference panel (Supplementary Table 17) provided higher accuracy than the 1KGP3 panel alone (Fig. 5C), particularly for *HLA-B* (an increase of 5.87%).

**Discussion**

São Paulo is the largest city in Latin America, with over 12 million individuals, and captures the Brazilian population's main structure. Since WGS will become the standard genomic tool for research purposes and the future of precision medicine, providing a reference for admixed populations is critical. Genomic datasets such as gnomAD and TOPMed have recently included Latin American samples, but this is the first study to include more than 1000 high-coverage WGS in any Latin American census-based cohort. Moreover, Brazil is not represented in these databases, although it is the only Latin American country colonized by Portugal and the destination of the largest contingent of individuals brought by the slave trade from the East, Central, and

West regions of Africa[47], and homeland of hundreds of Native American groups. During the 19th and 20th centuries, São Paulo was the destination of other Europeans (Italian, German, Dutch, Polish, Spanish), Middle Eastern (Syrians and Lebanese), and East Asian (Japanese) immigrants[20].

Even though the SABE sample size is modest compared to other initiatives, we have identified over 76 million short variants (SNVs and indels), of which ~2 million are absent from major public databases (Supplementary Fig. 4). We highlight that those elderly individuals unaffected by rare genetic disorders are useful controls and support pathogenicity classification. Regarding structural variation, we found a large set of ~2000 mobile element insertions not previously described and nearly 5 Mb of genomic segments absent from human genome reference (version GRCh38). Additionally, over 140 HLA alleles were inferred in our sample but not found in other databases. Whole-genome and HLA imputation were improved by the dataset when combined with 1KG3P, pointing that sample size can be, to some extent, compensated by diversity and representativeness. All results emphasize how WGS of admixed populations contribute as resourceful assets for population medical genomic studies, as well as for improving the human reference genome and the development of precision medicine.

## Methods

**Samples**. SABE is a census-based longitudinal study of elderly individuals that reside in the city of São Paulo, Brazil. Details on sampling and study design can be found in Supplementary Information and Supplementary Fig. 1. All subjects in the genomic dataset have agreed on participating in this study on written consent forms approved by CEP/CONEP (Brazilian local and national ethical committee boards).

**Sequencing and quality control**. Whole-genome sequencing was performed at Human Longevity Inc. following protocols previously described[4]. Library preparation was carried out using the TruSeq Nano DNA HT kit, and whole-genome sequencing was targeted at 30X and performed in Illumina HiSeqX sequencers using a 150 base paired-end single index read format. Reads were mapped to human reference GRCh38 using ISIS analysis software[4]. The sex of the samples was checked against proportions of reading pairs concordantly mapped to the X chromosome and male-specific part of Y chromosomes (MSY) related to those mapped to autosomes. As expected, females showed around 55,000 CPM X chromosomal reads and below 200 CPM, while genomic data from males showed these values being around 27,500 CPM and above 550 CPM, respectively.

Following GATK's Best Practices for germline short variant discovery (single nucleotide substitutions and insertion/deletions) and using GATK software (3.7 release)[48], we first generated individual GVCF (HaplotypeCaller) and then combined the GVCFs of all individuals (CombineGVCFs) to jointly call variants (GenotypeGVCFs) and perform Variant Quality Score Recalibration (VQSR-AS). Further, we used an in-house script to split the multiallelic variants into multiple lines and BCFtools[49] to standardize variants by left alignment. Annovar[50] and an in-house script were used to cross-reference the variants with dbSNP, 1000 Genomes Project, and gnomAD. The VEP-plugin LOFTEE[3] (v0.3-beta [https://github.com/konradjk/loftee]) was used to identify putative loss of function (pLOF) variants in at least one transcript irrespectively of confidence labeling.

We have previously developed an in-house two-step algorithm, CEGH-Filter, to evaluate the quality of called variants and genotypes[18], by directly flagging genotypes based on the depth of coverage and allele balance using hard cutoffs. Variants are flagged based on proportions of flagged genotypes, to provide insight into site-context batch effects (Supplementary Fig. 3). All analyses involving SNVs and indels resulted from filtering out GATK VQSR-AS non-PASS variants and lower confidence flags from the in-house CEGH-Filter v1 [https://github.com/cegh/SABE1171/blob/master/SABE_VCF_filter.pl] (Supplementary Information). A summarized table of computational steps, software, versions, packages, and datasets used throughout this article can be found in Supplementary Data 1.

Initial related analysis using KING[51] identified 28 pairs of relatives (sibships and duos), and only one individual from the pair was selected as proband by the following order of criteria: having brain MRI, oldest age, and being male. We used PC-Relate implemented in the GENESIS software[52] and the same dataset used for Admixture (see topic below) to confirm that no first degree relatives remained in the sample. verifyBAMID[53] identified one sample with over 3% of contamination, leading to its exclusion. A final dataset of 1171 unrelated participants was used in downstream analyses (Supplementary Fig. 1). Samples reached a minimum mean depth of coverage of 31.3× up to 64.8×, with an average depth of coverage of 38.65× and a median of 36.6× (Supplementary Fig. 2).

**Ancestry analyses**. We used ADMIXTURE v.1.3.0[54] to perform global ancestry inference through supervised analysis ($K = 4$) and 2000 bootstrap replicates, which uses a maximum likelihood framework, based on multilocus SNP genotypes. African (AFR, $N = 504$), European (EUR, $N = 503$), and East Asian (EAS, $N = 400$) non-admixed samples from 1KGP3[55], and Native Americans (NAM, $N = 221$) from recently published datasets[56], were used as parental populations (Supplementary Table 4). The Native American samples were genotyped on the Illumina Omni 2.5 M array; thus the genetic variants of the 1KGP3 and SABE samples (dataset of PASS (GATK) and vSR (CEGH-Filter, Supplementary Fig. 3) variants with genotypes flagged by CEGH-Filter as FD or FB set as missing) were filtered to overlap with this array, totaling 1,842,125 SNPs. LD-pruning on this subset of markers was performed with PLINK v.1.9[57], with an $r^2$ threshold of 0.1 within a sliding window of 50 Kb and a shift step of 10 Kb, resulting in 372,527 SNPs. We also used the same LD-pruned dataset to perform PCA analysis with R package SNPRelate[58] (Supplementary Fig. 5B). Our choice of parental populations to maximize ancestry inference accuracy was based on the tri-hybrid model (AFR, EUR, and NAM) previously used by other authors[42], which converges with historical evidence. We added EAS due to the high proportion of Asian immigrants (mainly from Japan) settled in São Paulo[20]. Before running ADMIXTURE in the SABE sample, we performed unsupervised runs using only parental populations to check if they formed distinct clusters and if there were any admixed individuals between them. In this way, we can use these contemporary individuals as a proxy for ancestral populations.

**Clinical analyses**. To evaluate the occurrence and clinical significance of pathogenic variants in genes associated with Mendelian disorders, a comprehensive panel containing 4250 OMIM disease genes (Supplementary Data 2) was retrieved and used for filtering SNVs and indels annotated with ClinVar pathogenic assertions (Pathogenic, Likely Pathogenic and Conflicting containing Pathogenic) and/or pLOFs identified by LOFTEE[3]. Classification of modes of inheritance was based initially on OMIM references, and upon manual curation with ClinGen[59] (https://clinicalgenome.org/) and PanelApp[60] (https://panelapp.genomicsengland.co.uk/). Manual curation was performed using ACMG recommendations[13,30], with current literature and evaluation of the most recent phenotypes collected in SABE follow-up, when available. The incidence of selected recessive disorders was calculated using the direct count of pathogenic alleles after manual curation and verification of pathogenicity in Locus Specific Databases (Supplementary Information 4.6). We have calculated the expected incidence based on the mode of inheritance pattern and assuming panmixia, all within cohorts (SABE versus gnomAD). Summary of steps and workflows can be found in Supplementary Information and Supplementary Figs. 4, 5.

**Mobile elements insertions**. Mobile Elements Insertions (MEIs) were detected using Mobile Element Locator Tool[61] (MELT; ver. 2.1.4). Specifically, MEIs (Alu, LINE-1, HERVs, and SVA) absent from the reference genome (GRCh38) were called with the MELT-SPLIT program and reference MEIs were genotyped using the MELT-Deletion program using the recommended standard calling procedures (https://melt.igs.umaryland.edu/manual.php). Next, additional filters were used to obtain a high-quality call and genotyping of MEIs. We filtered out (i) candidates not classified as "PASS" by MELT; (ii) candidates inserted in a low complexity genomic region; (iii) candidates presenting more than the expected number of discordant read pairs at the insertion site. For SABE-private and singletons events, we also applied additional filters. We selected only MEIs with MELT ASSESS score equal five, with a defined Target Site Duplication (TSD) domain and with minimal support (>2) split reads defining the insertion point. The assignment of LINE-1, Alu, and HERVs events to families and subfamilies was also performed using MELT. SVAs insertions were not subclassified in families.

MEIs (Alu, LINE-1, HERVs, and SVA) discovered among SABE samples were compared to MEIs present in the Database of Genomic Variation (DGV[36]), which includes Genome Aggregation Database (gnomAD) WGS samples. SABE events found in DGV were classified as Shared MEIs and are potentially polymorphic in humans. Only the same mobile element (e.g., Alu-Alu, L1-L1, HERV-HERV, or SVA-SVA) in the same genomic region was considered to be the same event, considering a ± 20 bp window of positional tolerance. Different classes of mobile elements falling in the same position are considered separate events. This overlap tolerance was based on the following possibilities: if there was a single ancestral event in the parental population followed by lineage-specific rearrangements, or calling discrepancies, or if there were independent events; regarding functional consequences and context interpretation, the overlapping events could be treated similarly. Manual examinations of the MEIs coordinate differences between our and public data revealed that the differences could be the result of variation in the TSD length or alignment adjustments.

To classify the genomic locations of MEI identified in the SABE genomes into genic (CDS, UTR, Intronic + flank) or intergenic, we matched the event coordinates against the GENCODE database. GENCODE (version 32) was used to define the set of transcribed regions. Exonic (CDS and UTR) and intronic regions (including 2k bp up and downstream the transcription start/end site) were defined as genic regions; all other genomic locations were defined as intergenic. In-house scripts were used to match MEIs coordinated to these regions aforementioned. In order to investigate the GC or AT composition of mobile elements insertion region,

first, we randomly selected 10,000 windows of length 100 bp from the human reference genome (GRCh38) and calculated their GC content (control). Second, we made the same for all mobile element insertion regions, discriminating by *Alu*, L1, SVA and HERV. Finally, we tested with the Kolmogorov-Smirnov test (KS test) the random windows distribution (control) against the distribution of mobile element insertion point.

**Non-reference nonredundant DNA segments library**. Unmapped (to GRCh38) paired reads from each individual were filtered for low-quality reads (average base quality below 20) and assembled using Megahit de novo assembler[62]. Non-reference sequence contigs (NRS) from the 1171 individuals were cross-assembled again with Megahit, and sequences longer than 200 bp were retained as non-redundant DNA segments. We aligned nonredundant segments against GRCh38 (including alternative haplotypes and decoy segments), using minimap[63], and we filtered out sequences with an identity of 95% or higher. We checked for bacterial and viral contaminations by blasting NRS against NCBI nonredundant database[64].

To determine the presence/absence of NRS in each individual, we aligned unmapped reads from each individual to GRCh38 extended with NRS, using bowtie2[65]. We discarded NRS for which none of the individuals showed read coverage in the range of 7.5–100× as potential contaminants or misassembled contigs. For coverage calculation, we considered only reads with mapping quality above 20.

Three sources of data were used for determining the genomic positions of NRS. (i) For contigs where the only part of it mapped to GRCh38, and the remaining portion (at least 200 bp) did not, the mapping coordinate of the former was used for anchoring the non-reference part of the contig to chromosomal location. (ii) Discordantly aligned read pairs (when mapped against GRCh38 + NRS) in which one read is aligned to NRS and its pair mate is aligned to a chromosomal location. (iii) We used publicly available 10× Chromium linked-reads data[66] from 26 Human Genome Diversity Project individuals (HGDP)[67] and nine Human Genome Structural Variation Consortium individuals (HGSVC)[68] to find overlap between barcodes mapped to NRS and chromosomal regions. Using bowtie2, we aligned 10× Chromium Genomes Linked Reads data to extended GRCh38 + NRS reference and extracted barcodes for reads uniquely mapped to NRS. The best target location for each NRS was defined as a location with the highest cross-sample number of linked reads with matching barcodes (per 1 kb window). NRS was considered as reliably localized if the best target location was discovered by at least two chromium barcodes (in the same or different individuals). Mapping reads to GRCh38 and GRCh38 + NRS was done bwa-mem mapper v.0.7.17[69]. Mapping positions of NRS anchored to mitochondrial DNA or decoy sequences were not reported. In cases when multiple mapping information was available, the preference was given to coordinates obtained by partial mapping or discordant paired reads giving more precise genomic coordinates. The outline of the NRS assembly pipeline steps and tools used are given in Supplementary Fig. 12.

**Whole-genome imputation**. To create the SABE reference panel, we used only variants flagged PASS (GATK) and vSR (CEGH filter), we set genotypes flagged as FD or FB (by CEGH filter) as missing and removed variants with >5% of missing genotypes. We used SHAPEIT2[70] to infer the chromosome phase using the extractPIRs tool, which incorporates the phase information contained in sequencing reads, improving phasing quality, particularly at rare variants[71]. We used the public 1000 Genomes Project Phase 3 haplotypes (1KGP3), version 27022019, including phased biallelic variants for 5248 unrelated samples, that were directly aligned against GRCh38[41]. The SABE + 1KGP3 reference panel was obtained by the merge of the SABE and 1KGP3 reference panels using the IMPUTE2 program[72].

To evaluate imputation performance, we used the EPIGEN-2.5 M dataset comprising 6,487 Brazilians from three population-based cohorts from Brazil genotyped on the Illumina Omni 2.5 M array:[13] (i) 1,309 children from Salvador with 51% of African, 43% of European, and 6% of Native American ancestry; (ii) 1442 elderly from Bambuí with 16% of African, 76% of European, and 8% of Native American ancestry; and (iii) 3736 young adults from Pelotas with 14% of African, 79% of European, and 7% of Native American ancestry. In addition, we evaluated imputation performance in two other admixed Latin American populations: (i) 391 Mestizos from Peru genotyped with Illumina Omni 2.5 M array[43], and (ii) 640 admixed individuals from Guatemala genotyped with Infinium OncoArray-500K BeadChip (unpublished dataset). We used CrossMap[73] to convert genome coordinates from hg19 to GRCh38 assembly, and removed SNPs with more than 5% missing.

We checked the consistency of the SNP's strand of the target and each reference panel with SHAPEIT2 using the human genome reference sequence GRCh38, and we used PLINK software[74] to flip the strands in case of inconsistencies. We phased the target EPIGEN-2.5 M, Peruvian and Guatemalan datasets using (1) the SABE haplotypes as phasing references, for the imputation with the SABE reference panel; (2) the 1KGP3 haplotypes as phasing references, for the imputation with the 1KGP3 reference panel; and (3) the 1KGP3 haplotypes as phasing references, for the imputation with the SABE + 1KGP3 reference panel.

We used IMPUTE2 to perform the imputation for chromosomes 15, 17, 20, and 22, on chromosome chunks of 7 Mb, with an additional 250 kb of buffer on both

sides (these were used for imputation inference but omitted from the results) and set the effective size parameter (Ne) to 20,000. We used IMPUTE2 info score as a metric of imputation quality, in which a value of 0 indicates that there is complete uncertainty about the imputed genotypes, and 1 indicates certainty about the genotypes.

To test imputation accuracy, we used the squared correlation ($r^2$) obtained by internal cross-validation performed by IMPUTE2. To this, IMPUTE2 masks the genotypes in the target panel, one by one, imputes the masked genotypes, and then compares the original genotypes with the imputed genotypes for each masked variant.

**HLA variants and haplotypes processing**. WGS reads from the SABE cohort were processed as described earlier. For the 1000 Genomes dataset, we obtained high-coverage BAM files using the ASPERA protocol. We processed these BAM files using hla-mapper version 4[44] [www.castelli-lab.net/apps/hla-mapper], as described elsewhere[45,75].

We used GATK HaplotypeCaller[48] version 4.1.7 to call genotypes in the genome confidence model (GVCF), concatenating all samples together in a VCF file using GenotypeGVCFs. We processed each HLA locus separately. For variant refinement and selection, we used the vcfx checkpl, checkad, and evidence algorithms to introduce missing alleles in genotypes with low likelihood and annotate each variant with a series of quantitative parameters[45] (www.castelli-lab.net/apps/vcfx). Each variant that has not been approved by the vcfx evidence algorithm was evaluated manually. The hla-mapper/GATK/vcfx workflow allowed the detection of 2257 high-quality variants considering 6 HLA class I loci, *HLA-A*, *HLA-B*, *HLA-C*, *HLA-G*, *HLA-E*, and *HLA-F*, against only 910 (40%) when using the regular workflow applied to the entire genome. We also calculated gene diversity, allele richness, and the mean number of different haplotypes across the 1000 Genomes populations and SABE using a local Perl script, resampling 50 samples in 5000 batches. This dataset was used in the analysis presented here.

For haplotype inference, we combined both physical phasings using GATK ReadBackedPhasing (RBP) and probabilistic models, as described in the supplementary material. After, we exported the phased data to complete sequences (exons + introns) and CDS sequences (only exons), comparing them with the ones described in the IPD-IMGT/HLA Database version 3.4.0[76]. Allele, genotype, and haplotype frequencies were calculated by direct counting. Please refer to the supplementary material for other details regarding the HLA workflow.

**HLA imputation**. Multi-ethnic imputation models for each of the class I classical HLA genes (*HLA-A*, *-B*, and *C*) were fitted using as reference panel: (a) SABE (1171 samples); (b) 1KGP3 (2503 samples); and (c) SABE + 1KGP3 (3674 samples). The imputation models were built with Attribute Bagging method implemented on HIBAG v.1.4[77], based on overlapping SNPs with the Axiom Human Origins array (Affymetrix), with HLA allelic resolution at the protein level (HLA—2 fields), 100 classifiers, and other default settings (Supplementary Table 25). To assess the accuracy of the models, imputation was performed on a sample of 146 highly admixed Brazilian individuals (43% AFR, 41% EUR, and 16% NAM) previously genotyped on Axiom Human Origins array and had HLA genotyped by standard methods (see details in Nunes et al., 2016[46]). To verify the accuracy of the imputation in each locus, the number of chromosomes with the correct HLA call was quantified over the total number of imputed chromosomes. The empirical cumulative distribution function (ECDF) was performed to access the posterior probability frequency distribution associated with the different reference panels (Supplementary Fig. 38)

**Reporting summary**. Further information on research design is available in the Nature Research Reporting Summary linked to this article.

## Data availability

The publicly available genomic dataset analyzed on this study is aggregated as a cohort and presented as short variants and frequencies deposited at the ABraOM [https://abraom.ib.usp.br], where they can be consulted and downloaded for academic research purposes via direct request at the website. ABraOM does not issue datasets with DOIs. Variants and frequencies were also submitted to dbSNP (to be published in the b156 release). Like the detection of short variants, class I HLA alleles and annotated mobile element insertions were detected using published software. Their lists of variants and respective frequencies are also available at ABraOM [https://abraom.ib.usp.br]. Imputation panels can be requested to corresponding authors. Individual-level sequence datasets (BAM files) and variant calling datasets (gVCF files) have been deposited at the European Genome-phenome Archive (EGA)[78], which is hosted by the EBI and the CRG, under EGA Study accession number EGAS00001005052. Further information about EGA can be found on https://ega-archive.org. All requests shall be made through EGA, to be evaluated and approved by the appointed Data Access Committee (DAC). SABE individual-level phenotypic data are not authorized by IRB to be uploaded to a public repository, although a direct collaboration is possible. Requests for phenotypic data use can be made directly through EGA, in which the DAC will evaluate each request. The timeframe for final approval is 180 days.

## Code availability

Most steps of the pipelines and datasets used on this study are publicly available and described within Methods and Supplementary Information are published and cited accordingly (Methods, Supplementary Data 1). Specific scripts developed to perform this study, such as quality assessment flags (CEGH-Filter) are detailed by an algorithm scheme (Methods, Supplementary Fig. 3); in-house codes are available at GitHub cegh/SABE1171. They can be obtained upon request until then.

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

## Acknowledgements

We acknowledge SABE participants for their long-term contribution and Prof. Maria Lúcia Lebrão (in memoriam) for her conceptualization and conduction of SABE. Funding was provided by FAPESP grants and fellowships (CEPID 2013/08028-1, SABE 2014/50649-6, INCT 2014//50931-3, 2013/17084-2, 2017/19223-0, 2012/24731-1, 2018/15579-8, 2015/25020-0, 2020/02413-4) and Conselho Nacional de Desenvolvimento Científico e Tecnológico (CNPq INCT 465355/2014-5). S.Z. and V.G. were supported by ALW Open grant ALWOP.662, financed by the Dutch Research Council (NWO). K.N. and D.M. were supported by United States National Institutes of Health—NIH (R01 GM075091). E.T-S group is funded by the Brazilian Ministry of Health—MoH/Brazil National Programme of Genomic and Precision Health—Genomes Brazil and Rede Mineira de Genomica Populacional e Medicina de Precisão (FAPEMIG). We thank the Human Longevity Inc. team that supported and conducted whole-genome sequencing. We acknowledge the staff from HUG-CELL and USP Public Health School. This study makes use of data generated by the Peruvian Genome Diversity Project (OI003-11 and OI-087-13) of the Peruvian National Institute of Health (data provided by C.P.R., C.S., and O.C.).

## Author contributions

M.S.N., M.O.S., and G.L.Y. conducted short variant analyses, managed ABraOM updates, and wrote the main manuscript. G.L.Y., J.Y.T.W., and M.O.S. performed and adapted the main bioinformatics pipeline. M.S.N., G.L.Y., and J.R.M.C. curated clinical variants. K.N. and D.M. performed ancestry analyses and human leukocyte antigen (HLA) genes imputation models. S.Z., T.K., and V.G. performed Non-Reference Sequence analyses. D.L.C., C.E.S.S., D.B., and R.N. developed and maintained ABraOM. M.O.S., N.M.A., W.C.S.M., and E.T.S. developed and performed WGS imputation pipeline. N.S.B.S., A.S.S., M.R.S.P., C.F.B.C., H.S.A., C.T.M.J., and E.C.C. developed the HLA pipeline and conducted the HLA variant and haplotype calls. R.L.V.M., J.L.B., F.O.R., T.L.A.M., and P.A.F.G. conducted mobile elements insertions (MEIs) analyses. R.C.M.N., V.B., H.G., O.C., C.P.R., C.S., M.D., E.T.S., B.L.H., M.L.B., and M.F.L.C. provided additional datasets used in imputation. E.T.S., D.M., V.G., P.A.F.G., and E.C.C. wrote sections of the main manuscript. M.S.N., M.O.S., G.L.Y., and M.Z. conceived the study. Y.A.O.D., M.R.P.B., and M.Z. provided SABE cohort samples, phenotypic data, and genomic data. All authors reviewed the manuscript.

## Competing interests

The authors declare no competing interests.

## Additional information

[1]Human Genome and Stem Cell Research Center, University of São Paulo, São Paulo, SP, Brazil. [2]Department of Genetics and Evolutionary Biology, Biosciences Institute, University of São Paulo, São Paulo, SP, Brazil. [3]Hospital Israelita Albert Einstein, São Paulo, SP, Brazil. [4]Instituto da Criança, Faculdade de Medicina da Universidade de São Paulo, São Paulo, SP, Brazil. [5]Orthopedic Research Labs, Boston Children's Hospital and Department of Genetics, Harvard Medical School, Boston, MA, USA. [6]Laboratório DASA, São Paulo, Brazil. [7]Laboratory of Genome Structure and Ageing, European Research Institute for the Biology of Ageing, University Medical Center Groningen, Groningen, Netherlands. [8]São Paulo State University (UNESP), Molecular Genetics and Bioinformatics Laboratory, School of Medicine, Botucatu, State of São Paulo, Brazil. [9]São Paulo State University (UNESP), Department of Pathology, School of Medicine, Botucatu, State of São Paulo, Brazil. [10]Departamento de Química, Faculdade de

Filosofia, Ciências e Letras de Ribeirão Preto, Universidade de São Paulo, Ribeirão Preto, São Paulo, Brazil. [11]Centro de Oncologia Molecular, Hospital Sirio-Libanes, São Paulo, Brazil. [12]Department of Biochemistry, Institute of Chemistry, University of São Paulo São Paulo, São Paulo, Brazil. [13]Bioinformatics Graduate program, University of São Paulo, São Paulo, Brazil. [14]Departamento de Genética, Ecologia e Evolução, Instituto de Ciências Biológicas, Universidade Federal de Minas Gerais, Belo Horizonte, MG, Brazil. [15]Núcleo de Ensino e Pesquisa, Instituto Mário Penna, Belo Horizonte, MG, Brazil. [16]Laboratorio de Biotecnologia y Biologia Molecular, Instituto Nacional de Salud, Lima, Peru. [17]Universidad de Huánuco, Huánuco, Peru. [18]Division of Cancer Epidemiology and Genetics, National Cancer Institute, Bethesda, MD, USA. [19]Instituto de Saúde Coletiva, Universidade Federal da Bahia, Salvador, BA 40110-040, Brazil. [20]Center for Data and Knowledge Integration for Health, Institute Gonçalo Muniz, Fundação Oswaldo Cruz, Salvador, BA, Brazil. [21]Instituto de Pesquisas René Rachou, Fundação Oswaldo Cruz, Belo Horizonte, MG, Brazil. [22]Programa De Pós-Graduação em Saúde Pública, Universidade Federal de Minas Gerais, Belo Horizonte, MG, Brazil. [23]Programa de Pós-Graduação em Epidemiologia, Universidade Federal de Pelotas, Pelotas, RS, Brazil. [24]Mosaico Translational Genomics Initiative, Universidade Federal de Minas Gerais, Belo Horizonte, MG 31270-901, Brazil. [25]Facultad de Salud Pública y Administración, Universidad Peruana Cayetano Heredia, Lima, Peru. [26]Instituto de Estudos Avançados Transdisciplinares, Universidade Federal de Minas Gerais, Belo Horizonte, MG 31270-901, Brazil. [27]Medical-Surgical Nursing Department, School of Nursing, University of São Paulo, São Paulo, SP, Brazil. [28]Epidemiology Department, Public Health School, University of São Paulo, São Paulo, SP, Brazil. [29]These authors contributed equally: Michel S. Naslavsky, Marilia O. Scliar, Guilherme L. Yamamoto. ✉email: mnaslavsky@usp.br; mayazatz@usp.br

