## [Peer Review File · Nature Communications]

Title: Whole-genome sequencing of 1,171 elderly admixed individuals from São Paulo, BrazilREVIEWER COMMENTS

Reviewer #1 (Remarks to the Author):

Summary

=====

The authors present a cohort sequencing study involving short read WGS of over 1000 elderly Brazilian individuals. WGS data are analysed to reveal patterns of ethnicity, mobile element insertions (MEIs), and non-reference sequences, and the presence of both rare and common variation is correlated to participant phenotypes. The authors also apply their data to build genomics tools, in the form of an improved imputation panel for South American individuals, and a database of HLA alleles.

The authors argue convincingly for the importance of their dataset, which although small, captures variation in an understudied genomic population. Overall, I find this is a well-executed study that presents a valuable dataset of interest to the genomics and clinical genomics communities. The authors' findings for 'clinically relevant' variants in particular could be of great relevance to the clinical genetics field, if its conclusions are strengthened through additional work. That said, I do have some concerns about data availability and the GWAS.

Major concerns

=====

1. Most notably, the data sharing arrangement does not meet the contemporary standard. *At a minimum*, raw data should be deposited in a central repository designed for this purpose (eg EGA), to provide redundancy and ease sharing. This is absolutely essential for a resource such as the SABE, which needs to be accessible for its benefits to be fully realised. Ideally, a formal institutional data access committee should also be established, to help ensure that data will remain available beyond the tenure or interest of the corresponding authors.

2. I have some concerns about the GWAS. For example, the cancer GWAS found one significant locus, in females only, with an OR of 30, on the basis of fewer than 77 cases. This is not mathematically impossible, but is striking, especially considering that genomic region has not been reported to be associated with cancer. It is also slightly worrying that relatively few of the hits from the full WGS loci GWAS were also detected in the Omni2.5M-loci-only GWAS; this does make me wonder if the full WGS loci set contains loci that are especially susceptible to technical biases, which are less common in the Omni2.5M loci which focuses on 'easier' genomic regions. I have observed an effect similar to this in my own work, so it is not pure speculation. At any rate, my suggestion is either:

- a) Perform validation for this section, ideally with a genuine SNP array validation cohort, or
- b) Remove this section, as it's not clear to me that it adds much value to the manuscript.

Minor concerns

=====

* "Privative" has been consistently used when I think the authors mean "Private".

* In the MEI section the term "lineage-specific germinative MEI" is used. I am not an expert in this area, but wonder if the authors are perhaps describing germline MEIs private to one individual in the cohort. If so, a different term from "lineage-specific germinative MEI" would probably be less confusing to most readers.

Comments on improving impact

=====

1. In my opinion, the most impactful part of this manuscript is the "Clinically Relevant Findings" section. In this section, 31% of rare variants in the SABE WGS that had a P or LP annotation in ClinVar could be reclassified as VUS/B, on the basis of manual review. This is a striking finding that could influence thinking in the field of clinical genetics, and really underscore the value of the SABE WGS cohort, but it is barely considered in the text. I suggest exploring this further would increase interest to a clinical genetics audience:

* Exactly what was the process that led to the reclassification? Were appropriately qualified individuals involved (eg genetic pathologist), and how were decisions made? This is touched upon in the supplementary methods but there isn't enough information available to recapitulate the process used. (Can the reclassified & retained variants be shared as a supplementary table, along with the comments/annotations leading to their reclassification? Something akin to supp table 11, but for all OMIM ClinVar P/LP variants. That would help clarify the approach.)

* What was the composition of ClinVar confidence levels in the 31%? Were they all conflicting or supported by only one entry, or were some of them of higher confidence? Were they largely old / literature only entries, or were some more modern?

* Is this high rate of incorrect ClinVar P/LP annotations a general aspect of ClinVar variants, or do the SABE WGS variants tend to have false P/LP ClinVar annotations, as they are relatively rare in the patients whose variants have contributed to ClinVar? I suspect the latter, and if this is the case, it really underscores the importance of healthy cohorts of broad population background to establish variant pathogenicity.

* Overall, this section was a little difficult to follow and needed frequent reference to the supplement to fully understand it; a rewrite may make its importance more clear.

2. Related to the above, the authors describe MEIs and even mention a high rate in genic regions (including introns). Have the authors analysed these MEIs to determine if any disrupt clinically relevant genes? This would be an interesting finding with relevance to rare disease genetics, and should be little work as the authors already have all the necessary analyses done.

3. It would be very interesting to see an analysis of SV or CNV in this cohort, but I recognise that this would be a significant undertaking perhaps best left to another manuscript.

4. Does remapping your data with a custom reference that includes your NRS improve variant calling

accuracy, for example by reducing the number of CEGH filtered loci, or by reducing false positive MEI/SV/CNV calls?

Mark Pinese

Reviewer #2 (Remarks to the Author):

Naslavsky et al. present the whole-genome sequencing from 1,171 elderly admixed individuals from Brazil. The data set presented by the authors contributes to filling the gaps in the genetic variation/diversity of Latin American populations.

In general, the paper is highly descriptive. Authors should discuss their results by integrating previously published data sets for Latin America, such as those of Harris et al.

(<https://doi.org/10.1073/pnas.1720798115>). Furthermore, the manuscript could benefit from a more detailed description of the methods used for the various analyses. For example, specific analysis for determining the incidence of monogenic diseases and the GWAS are not specified and this makes it difficult to evaluate these results.

I would like the authors to include the following suggestions in a revised version.

Line 97-99: How unaffected elderly individuals can help to improve diagnosis accuracy of rare diseases? In my opinion, the NGS can help to improve diagnosis and the use of large datasets of unaffected individuals helps to estimate the prevalence of these diseases in specific populations.

Line 104: Sentence “immigrant descendants from different continents and individuals from various Brazilian states” has to be clarified. It is unclear if all participants are Latin Americans or some of them are from different continents living in Brazil.

Line 106: Please cite a reference for SABE study.

Line 122: Please add a brief description of the pipeline used to analyze the 1171 genomes, a measure of dispersion should be added to the average data.

Line 125: According to section methods the authors annotate the variants with Annovar, the authors could be more explicit about the number and the consequences of the variants identified in this study (missense/silent, LOF, frameshift, etc).

Line 129: Change global ancestries by genetic structure. In the corresponding section the authors should discuss how individuals with ancestry mainly European and African (and with a low proportion of Native American ancestry) can represent the Latin American population.

Line 129-134: Could the authors add a description of the method used to determine the genetic

structure of your cohort. Are relevant the PCA plot presented in Fig 1B? Also, a description of how was calculated is needed.

Line 134-137: This sentence is no clear.

Figure 1: Please include the parental populations used as panel for the Admixture K= 4. Why the sample size for SABE cohort changed? Please justify. Except for Asian ancestry, it is difficult to know what ancestry are representing the colors. In general, for panels A & B, I suggest to use contrasting colors. Authors should be particularly careful using those labels in Fig1A.

Line 164: I recognize the hard-manual curation work, so a diagram or more details about this process should be added. It would also be relevant to compare these results with those derived from automated annotators such as InterVar ([10.1016/j.ajhg.2017.01.004](https://doi.org/10.1016/j.ajhg.2017.01.004)).

Line 174-178: The authors claim about incidence estimation for autosomal recessive diseases using population-based genetic data, nevertheless it is hard to know how these values were obtained due the lack of a description of the method used to obtain the incidence, are you following the pipeline reported by Bainbridge M? (<https://doi.org/10.1007/s00439-020-02135-5>). I encourage the authors to include a method section describing the methodological procedures for these estimations and to show the values of the incidence estimated in the main text.

Would it not have been useful to have family-based data to track for the transmission of low frequency causal variants in pedigrees to see if any segregate with disease?

Line 224: Change indicating by suggesting.

Line 263-264: Please add a brief description about the method used to perform de novo assembly.

Figure 4.- Please specify the letters for each panel

Figure 5.- The same comment as figure 4

Line 369-380. The GWAS story is hard to understand. In general, this section need to be improved. They performed GWAS for selected phenotype (BMI, LDL, triglycerides, positive history of cancer, cognitive decline, diabetes, frailty, and hypertension). Could be more explicit on the phenotypes used in GWAS? What about the possibility of undiagnosed cases among controls? Please mention the cutoffs to define the phenotypes. There is no mention of the statistical power. If the characteristics of the 1171 individuals were stratified by case or control, and then by gender and for multiple comparisons, is there an adequate statistical power? Authors should add a table or regional plots for the new 12 hits. Also, expand the discussion about the possible mechanism or biological processes in which they participate. I suggest to include an enrichment pathway analyses for the new and previously described hits. Finally, to address the association of rare variants with the studied phenotypes I suggests a sequence Kernel analysis.

Line 382-406. In general, the discussion does not add much to the previously discussed in other sections.

Line 519-758. In the methods section please add a section about the estimation of incidence of rare diseases.

Line 565. Based on what criteria the authors chose the reference population panel for admixture. The authors have to be cautious about supervised analysis due it is only suitable when the reference individuals can be assigned to ancestral populations with certainty and ancestral populations are fairly homogeneous. In this case, several populations from the 1KGP3, shows gene flow between them, causing that the ancestry fraction estimates in target dataset can suffer from bias.

Line 576. Did the authors use the PCAs calculated with the reference panel for ancestry correction in GWAS analyses? If true, you need to perform a new PCA estimations without the reference panel and use it to perform the ancestry adjustment.

Line 680. Change SNP's by SNVs.

In general, change SNP by SNV in the full text.

Reviewer #3 (Remarks to the Author):

In their manuscript, Naslavsky et al. present high-coverage WGS data for over 1100 elderly, Brazilian individuals. With these data, they identify ~2M genetic variants absent from other large public databases, including novel mobile elements, nonreference sequences, and over 140 novel alleles from HLA genes. They also reassess pathogenicity of variants associated with Mendelian disorders, calculate the expected incidence for selected recessive disorders in the Brazilian population, provide an improved panel for whole-genome and HLA imputation, and perform GWAS for ~10 traits. These findings are of potential interests for those in the fields of human genetics, genomics, population genetics, and anthropology, among others. I found all statistical methods to be valid and appropriate.

In my opinion, the strength of their work lies on the generation of a dataset comprising high-coverage WGS for over 1100 individuals from an underrepresented population in genomics, with associated data on medically-relevant phenotypes. As the authors note, currently available public genomic datasets lack non-European and admixed populations and this lack of diversity may lead to biases in precision medicine and consequently less accurate genetic tests on underrepresented populations, potentially increasing health disparities and affecting historically marginalized communities. I believe this study is

very timely and its dataset is an interesting resource for research in human genetics and genomics.

Please find additional comments below. Thank you!

Best regards,

C. Eduardo Guerra Amorim

Minor comments:

- a. Lines 95-96: Perhaps it would be good to cite some references from the literature that also discuss this relationship between expected and observed frequencies, disease prevalence, etc.
- b. Data availability: It reads on lines 113-114 that variants and allele frequencies are publicly available. I wonder whether this journal requires the full dataset (fastq, VCF or BAM files) to be available.
- c. Lines 134-137: What does partially account for ancestry variation? Is that the self-reported ethnoracial group? Can you briefly explain the model? Preferably in the main text or perhaps in Extended Data Figure 2 title? (Please check for typos e.g. “withing”)
- d. Line 159: missing word after “curated” (?)
- e. Lines 162-163: Do you mean you could only confirm the effect of 3% of the studied mutations? This number is surprisingly small! This seems to be an interesting result. Have previous studies also examined that for Mendelian diseases? Please add some references in the main text to help readers assess the importance of these findings.
- f. Paragraph starting in line 174: How do you calculate the expected incidence? Is that simply the number of homozygotes and compound heterozygotes? Or the number of expected offspring? Or just that the allele frequencies are similar between datasets?
- g. Lines 196-199: Are all these polymorphic in humans?
- h. Figure 5: Correct me if I am wrong, but I believe “America” should be replaced for “South America” or “Americas.”
- i. Can you give some reasoning for why relatively common variants (allele frequency >2%) are more often missing from other datasets than less common variants (AF 0.5% - 2%), but not very rare ones (AF <0.5%)?
- j. Line 568 (Methods): reference 10 seems to be focused on African ancestry. Please check if this reference is correct.

k. Last, I think the paper is generally well written, but some sentences are wordy and confusing. Below I show some examples:

(1) Lines 88-89: "Most importantly ... harbor specific variants."

(2) Lines 95-96: The frequency of what? The observed allele frequency? Do you mean the observed allele frequency cannot be larger than the expected frequency calculated based on disease incidence? Is this true for both recessive and dominant diseases? Would incomplete penetrance, complementation, advantage of the heterozygote (in recessive disorders), and late disease onset affect this relationship between expected and observed frequencies?

(3) Lines 159-163: What do you mean with curation has led to reclassification of pathogenicity *by* inheritance mechanism or penetrance?

(4) Lines 164-165: "Manual curation promotes ... to databases."

(5) Lines 171-173: "Also, regarding P/LP ... elderly cohort."

Responses to Reviewer comments

We have addressed the Reviewer's comments below. Each response is numbered and whenever sentences are added or altered in the Main text, Methods or Supplementary Information, we have (1) tracked changes to the submitted files (except changes in table renumberings, which were applied), (2) copied the context of the alterations and pointed the lines of the original manuscript to this file, and (3) highlighted the editions in bold/red font.

Reviewer #1 (Remarks to the Author):

Summary

=====

The authors present a cohort sequencing study involving short read WGS of over 1000 elderly Brazilian individuals. WGS data are analysed to reveal patterns of ethnicity, mobile element insertions (MEIs), and non-reference sequences, and the presence of both rare and common variation is correlated to participant phenotypes. The authors also apply their data to build genomics tools, in the form of an improved imputation panel for South American individuals, and a database of HLA alleles.

The authors argue convincingly for the importance of their dataset, which although small, captures variation in an understudied genomic population. Overall, I find this is a well-executed study that presents a valuable dataset of interest to the genomics and clinical genomics communities. The authors' findings for 'clinically relevant' variants in particular could be of great relevance to the clinical genetics field, if its conclusions are strengthened through additional work. That said, I do have some concerns about data availability and the GWAS.

Major concerns

=====

1. Most notably, the data sharing arrangement does not meet the contemporary standard. *At a minimum*, raw data should be deposited in a central repository designed for this purpose (eg EGA), to provide redundancy and ease sharing. This is absolutely essential for a resource such as the SABE, which needs to be accessible for its benefits to be fully realised. Ideally, a formal institutional data access committee should also be established, to help ensure that data will remain available beyond the tenure or interest of the corresponding authors.

R1) Thank you for your comments and suggestions. We completely agree that the deposit of raw data in a central repository benefits all the scientific community. We decided to submit raw data (individual level BAM and gVCF files) to EGA, accession number EGAS00001005052, where it can be retrieved upon request after approval of a Data Access Committee. Datasets are currently being submitted and uploaded under the registered study above. We expect that it will be available before publication. We added this information in the Data availability section (below, in red). In addition, we have included information on sharing the reference panel with haplotypes for imputation.

Actions to R1:

Removal of sentence in lines 767-769:

"Their lists of variants and respective frequencies are also available at ABraOM. Individual level genomic data (VCFs and BAM files) and phenotypic data can be shared upon reasonable request

to corresponding authors via collaboration and data use agreements approved by competent parties”.

Lines 767-769 now read:

“Their lists of variants and respective frequencies are also available at ABraOM. **Imputation panels can be requested to corresponding authors. Individual level sequence datasets (BAM files) and variant calling datasets (gVCF files) have been deposited at the European Genome-phenome Archive (EGA)[Lappalainen et al, 2015], which is hosted by the EBI and the CRG, under accession number EGAS00001005052. Further information about EGA can be found on <https://ega-archive.org>.** Phenotypic data can be shared upon reasonable request to corresponding authors via collaboration and data use agreements approved by competent parties.”

2. I have some concerns about the GWAS. For example, the cancer GWAS found one significant locus, in females only, with an OR of 30, on the basis of fewer than 77 cases. This is not mathematically impossible, but is striking, especially considering that genomic region has not been reported to be associated with cancer. It is also slightly worrying that relatively few of the hits from the full WGS loci GWAS were also detected in the Omni2.5M-loci-only GWAS; this does make me wonder if the full WGS loci set contains loci that are especially susceptible to technical biases, which are less common in the Omni2.5M loci which focuses on 'easier' genomic regions. I have observed an effect similar to this in my own work, so it is not pure speculation. At any rate, my suggestion is either:

- a) Perform validation for this section, ideally with a genuine SNP array validation cohort, or**
- b) Remove this section, as it's not clear to me that it adds much value to the manuscript.**

R2) The main idea of including preliminary GWAS findings in the current manuscript was to demonstrate the effect of including rare variation present in WGS-based on overall signal distributions in a GWAS setup compared to traditional array-based analyses. However, we agree with you and the other reviewers, that it needs further validation. Thus, we decided to remove the GWAS section from this manuscript. In order to provide appropriate validation and discussion on findings, a future manuscript is planned to include GWAS and burden/rare variant collapsing analyses, including downstream analyses such as replication and functional annotation of findings.

Actions to R2:

R2a) Removal of sentence in line 111:

“Additionally, we explored pathogenicity assertions in disease-related genes of clinical relevance and GWAS performance for selected phenotypes”.

Now reads:

“Additionally, we explored pathogenicity assertions in disease-related genes of clinical relevance”.

R2b) Removal of sentence in line 576:

“The PCs obtained were further used for ancestry adjustment in GWAS analysis.”

R2c) Removal of GWAS Section entirely. From lines 369 to 380 in the Main text and 738 to 758 in the Methods section.

R2d) Removal of Supplementary Tables 20-23.

R2e) Removal of Supplementary Figures 24-31.

Minor concerns

=====

* "Privative" has been consistently used when I think the authors mean "Private".

R3) Thank you for this observation. We have replaced 'Privative' for 'Private' in all occurrences (lines 201, 205, 211, 214, 247, and 248). Update was also applied to Figure 2 and legend.

* In the MEI section the term "lineage-specific germinative MEI" is used. I am not an expert in this area, but wonder if the authors are perhaps describing germline MEIs private to one individual in the cohort. If so, a different term from "lineage-specific germinative MEI" would probably be less confusing to most readers.

R4) The term was used to indicate uncertainty regarding the source of the variant (if somatic or germinative) and the frequency (it is a singleton in our cohort, but it might be shared with the relatives of the carrier's lineage). We agree it can be confusing, so we have replaced 'lineage-specific germinative MEI' by '**germinative MEI**' (line 207).

Comments on improving impact

=====

1. In my opinion, the most impactful part of this manuscript is the "Clinically Relevant Findings" section. In this section, 31% of rare variants in the SABE WGS that had a P or LP annotation in ClinVar could be reclassified as VUS/B, on the basis of manual review. This is a striking finding that could influence thinking in the field of clinical genetics, and really underscore the value of the SABE WGS cohort, but it is barely considered in the text. I suggest exploring this further would increase interest to a clinical genetics audience:

* Exactly what was the process that led to the reclassification? Were appropriately qualified individuals involved (eg genetic pathologist), and how were decisions made? This is touched upon in the supplementary methods but there isn't enough information available to recapitulate the process used. (Can the reclassified & retained variants be shared as a supplementary table, along with the comments/annotations leading to their reclassification? Something akin to supp table 11, but for all OMIM ClinVar P/LP variants. That would help clarify the approach.)

* What was the composition of ClinVar confidence levels in the 31%? Were they all conflicting or supported by only one entry, or were some of them of higher confidence? Were they largely old / literature only entries, or were some more modern?

R5) Thank you for your comments. Indeed we state that population-based elderly individuals are a very useful sample for (re)classification of variant pathogenicity. Two independent medical geneticists (GLY; JRMC) applied ACMG 2015 criteria with ClinGen modifications taking also into consideration genotype-phenotype correlations clinically known and reported in databases such as OMIM and literature review especially for reports of functional studies.

In addition to submitting these variants as Supplementary Tables (Supplementary Table 11, for variants that were reclassified and Supplementary Table 12, for variants that fell into other categories), we have decided to submit reclassified variants to ClinVar providing criteria. This

revision led to a slight reduction in the number of reclassified variants, from 123 (31%) to 116 (29%). Among the 116/394 (29%) reclassified, most were indeed originally submitted to ClinVar without criteria (70 variants). We have not analyzed the date of entries, but empirically we confirm that most reclassified variants had older assertions. We added the sentences (in red) below in the Main text and in Supplementary Information.

Actions to R5:

R5a) Lines 156-160 now read:

We analyzed 'Pathogenic' or 'Likely Pathogenic' (P/LP) ClinVar asserted variants carried by SABE individuals across 4,250 genes associated with monogenic disorders (Online Mendelian Inheritance in Man - OMIM disease genes, Supplementary Table 6) and manually curated **the variants in genes associated with dominant inheritance using ACMG guidelines²⁰ and literature support, performed by clinical geneticists (complete workflow on filtering, annotation, and counts are described in Supplementary Fig. 4 and Supplementary Information 4).** In total, out of 394 variants asserted as either P/LP in genes annotated to have at least one phenotype with a dominant inheritance, curation resulted in the reclassification **due to pathogenicity downgrade (116 or 29% of variants, Supplementary Table 11), or compatible categories with unaffected phenotypes due to inheritance mechanism (53%) or incomplete penetrance (143%), with only 3% of variants associated with a matching detectable phenotype (Extended Data Tab. 1, Supplementary Table 12).**

R5b) Supplementary Information 4.4. now read:

In order to identify individuals carrying variants with potential clinical implications, including the reassessment of related phenotypes to support the analyses, we have filtered a total of 394 variants asserted as either 'Pathogenic' or 'Likely Pathogenic' (P/LP) in genes annotated to have a dominant mode of inheritance only, and in genes with more than one mode of inheritance, including dominant or monoallelic. Manual curation aiming reclassification of pathogenicity using ACMG criteria was performed **by two independent clinical geneticists (professionals with clinical genetics residency and previous experience in clinical exome analysis and variant pathogenicity classification using ACMG criteria).** Manual curation included **functional studies and segregation information described in the available literature, evidence details on the original assertions, and allele frequency. Each of the 394 variants in dominant genes containing P/LP ClinVar assertions was submitted to manual curation aided by population-specific frequencies (gnomAD and SABE, mainly); ClinGen (to reannotate inheritance modes); review of VarSome automated calculation of ACMG classification criteria; and in-depth analyses on ClinVar submissions leading to classification, particularly in evidence levels (ACMG criteria assigned and provided by submitters, to adjust PP5), details on co-segregation of ClinVar assertion combined with literature reports of carriers and families (to adjust PP1). OMIM aided reclassification of gene's mode of inheritance in cases where ClinGen information could not be conclusive, such as only one affected case was reported and recessive mode could not be excluded. When loss of function consequence would only be detected in trans with another P/LP variant and not by itself (hypomorphic variants) the allele did not meet criteria for haploinsufficiency and dominant phenotype (hereby classified as 'recessive allele').** A total of **116** variants (29%) were reclassified as non-pathogenic assertions (benign, likely benign or unknown significance) (**Supplementary Table 11**), **most of which had no assertion criteria provided (70 variants), 44 had criteria provided by a single submitter and 9 by multiple submitters.** The remaining **278** kept as pathogenic or likely pathogenic (**Supplementary Table 12**). Among the latter, literature validation and matching phenotypes, when available, enabled further characterization of variants to either a reported reduced penetrance, non-dominant mode (of the specific allele or gene), or associated to clinical

features that are not severe enough to cause mortality before the average age of subjects (Extended Data Tab. 1).”

Supplementary Table 11. Reclassified variants after manual curation in genes with dominant mode of inheritance.

Large table displayed only on Supplementary Tables worksheet.

Supplementary Table 12. Pathogenic variants and categories after manual curation in genes with dominant mode of inheritance.

Large table displayed only on Supplementary Tables worksheet.

R5c) Line 164 now reads:

Manual curation promotes the downgrading of **P/LP** assertions when **larger sample sizes** and diverse ancestries are added to databases, **which may increase the frequency of some variants, allowing updates of older assertions which are likely to have an inflated proportion of P/LP assertions**[Kessler et al, 2016; Xiang et al, 2020].

*** Is this high rate of incorrect ClinVar P/LP annotations a general aspect of ClinVar variants, or do the SABE WGS variants tend to have false P/LP ClinVar annotations, as they are relatively rare in the patients whose variants have contributed to ClinVar? I suspect the latter, and if this is the case, it really underscores the importance of healthy cohorts of broad population background to establish variant pathogenicity.**

R6) Yes, we also had the same thought, as some variants were downgraded to VUS or B/LB, and most were flagged as lower penetrance, we have explored the correlation with global and local ancestries. The results are described in a different manuscript which was recently submitted. This correlation was previously observed by Kessler and colleagues (2016) who quantified disparities in ClinVar’s and HGMD’s misclassified assertions correlated to African ancestry. This reference was added to the above mentioned comment on the Main text.

*** Overall, this section was a little difficult to follow and needed frequent reference to the supplement to fully understand it; a rewrite may make its importance more clear.**

R7) We have rewritten parts of the section on Main text and Supplementary Information to clarify the workflow and highlight its importance (see Responses 5a, 5b, and 5c).

2. Related to the above, the authors describe MEIs and even mention a high rate in genic regions (including introns). Have the authors analysed these MEIs to determine if any disrupt clinically relevant genes? This would be an interesting finding with relevance to rare disease genetics, and should be little work as the authors already have all the necessary analyses done.

R8) We thank the reviewer for this valuable comment. As suggested, we investigated the insertions of MEIs into genic regions, both introns and exons, and provided specific counts on events occurring in all genes and in clinically relevant genes (OMIM Disease genes, Supplementary Table 6). In summary, we identified 107 exonic events, of which 26 are exclusive to SABE dataset (singletons + private) in exonic regions (added Supplementary Table 15). 20 events overlapped OMIM Disease genes (added Supplementary Table 16). A brief analysis was added to the main text, as presented below.

Actions to R8:

- 1) We briefly reported these findings to the 'MOBILE ELEMENTS INSERTIONS (MEIs)' section in the modified manuscript version:

Even though 26 events fall within exonic regions of clinically relevant genes (Supplementary Tables 15-16), none are described to have phenotypic manifestations in one copy or with loss of function mechanism (Supplementary Information).

- 2) We added to the Supplementary Information the following section and comments:

As presented in the main text and methods, mobile element insertions (MEIs) were identified across all samples and annotated by element types and frequency groups. Next, we have annotated the overlaps between occurrence of MEI events in OMIM disease genes (Supplementary Table 6) and their respective genomic contexts. Results can be found in Supplementary Table 15 (total counts) and Supplementary Table 16 (MEI events in exonic regions of OMIM Disease genes).

Supplementary Table 15. Counts of mobile element insertions per frequency group, genomic context and OMIM annotation

Supplementary Table 16. OMIM genes with mobile element insertion events in exonic regions.

Large table displayed only on Supplementary Tables worksheet.

Regarding MEIs identified in exonic regions of OMIM genes (Supplementary Table 16), all genes but *HCN1*, *PACS1* and *PIK3R1* are associated with recessive disorders, susceptibility loci or non-disease traits. *HCN1* variants associated with AD epilepsy are all missense with gain of channel function or dominant negative effects even though pLI in gnomAD is 1, multiple controls in Developmental Delay Database (DDD) have been identified with intragenic exon spanning deletions. Schuurs-Hoejmakers is associated with a single recurrent variant in *PACS1* (NM_018026.4: c.607C>T), and even though pLI in gnomAD is also 1, loss of function variants (or deletions) in this gene have never been reported associated with disease in humans. *PIK3R1* variants have been associated both with AR inheritance (loss of function, nonsense, variant) and AD (splicing, predicted gain of function, variants), pLI in gnomAD is 0.02. Therefore there is evidence that none of the MEI in OMIM genes that could potentially lead to truncation of the gene product (and a loss of function consequence) are likely to be associated with a severe disease phenotype in the individuals from SABE cohort.

3. It would be very interesting to see an analysis of SV or CNV in this cohort, but I recognise that this would be a significant undertaking perhaps best left to another manuscript.

R9) We agree and we are currently working in SV and CNV calling and annotation to be published in a different manuscript.

4. Does remapping your data with a custom reference that includes your NRS improve variant calling accuracy, for example by reducing the number of CEGH filtered loci, or by reducing false positive MEI/SV/CNV calls?

R10) We agree and we intend to explore a reference genome build “GRCh38+SABE_NRS” regarding variant calling gains and a correlation of CEGH-Filter flags and false negatives (due to reference effects). Although we have not called variants using the reference build with SABE-NRS due to computational and time constraints, we have mapped three randomly selected samples and compared results of mapping against GRCh38 and GRCh38 appended with SABE non-reference segments (GRCh38+SABE_NRS). In both cases we used bwa-mem mapper v.0.7.17 (Li and Durbin, 2009). Expectedly, aligning against extended reference decreased number of unmapped reads by 2.6% (range from 2.4-2.8%). Further, the primary alignments of the reads against GRCh38+SABE_NRS show increased proportion of mapped bases, by 0.3% (0.2-0.3%) and decreased number of soft-masked (8.4%, 7.0-9.8%), deleted (9.9%, 8.4-11.0%), and inserted bases (34.4%, 29.1-39.6%). This indicates that extended reference allows assigning additional reads and bases to the reference genome, while reducing the number of misalignments.

Actions to R10) We have inserted these results in line 319 of the main manuscript file

We have mapped three randomly selected samples and compared results of mapping against GRCh38 and GRCh38 appended with SABE non-reference segments (GRCh38+SABE_NRS). Expectedly, aligning against extended reference decreased number of unmapped reads by 2.6% (range from 2.4-2.8%). Further, the primary alignments of the reads against GRCh38+SABE_NRS show increased proportion of mapped bases, by 0.3% (0.2-0.3%) and decreased number of soft-masked (8.4%, 7.0-9.8%), deleted (9.9%, 8.4-11.0%), and inserted bases (34.4%, 29.1-39.6%), indicating that an extended reference reduce the number of misalignments. Additional investigation is required, including calling variants in Brazilian samples using the GRCh38 reference appended with SABE.

In the methods section (line 730):

Mapping reads to GRCh38 and GRCh38+NRS was done bwa-mem mapper v.0.7.17[Li and Durbin, 2009].

Reviewer #2 (Remarks to the Author):

Naslavsky et al. present the whole-genome sequencing from 1,171 elderly admixed individuals from Brazil. The data set presented by the authors contributes to filling the gaps in the genetic variation/diversity of Latin American populations.

In general, the paper is highly descriptive. Authors should discuss their results by integrating previously published data sets for Latin America, such as those of Harris et al. (<https://doi.org/10.1073/pnas.1720798115>). Furthermore, the manuscript could benefit from a more detailed description of the methods used for the various analyses. For example, specific analysis for determining the incidence of monogenic diseases and the GWAS are not specified and this makes it difficult to evaluate these results.

R11) Thank you for your comments. We agree with the reviewer that the data we present has the potential to address issues related to the history and demography of Latin American populations. However, in preparing the manuscript, we decided that a more coherent message would be achieved if we focused on a central theme, and for this reason we prioritized reporting novel

variation and interpreting the clinical implications of these findings. We therefore only present results for ancestry which give the broad context of the data but have left the historical interpretations to be treated elsewhere. Regarding the suggestion of a more detailed description of the methods, we added information for the manual curation (R5) and the determination of incidence, as described below and in Response 22. Also, we have decided to remove the GWAS section from this manuscript, as explained in R2.

Actions to R11) Regarding the determination of incidence, we have added the following sentences (in red) to item 4.6 of Supplementary Information:

“To roughly estimate the incidence using counts of heterozygotes from SABE and gnomAD global and population-specific datasets, we selected five genes associated with prevalent monogenic clinical phenotypes: cystic fibrosis (*CFTR*), hemoglobinopathies (*HBB*), deafness (*GJB2*), familial Mediterranean fever (*MEFV*), and hemochromatosis (*HFE*) (Supplementary Table 12). **These genes were used to filter high frequency (up to 5%) and low frequency (including singletons) known pathogenic variants, as classified by respective Locus Specific Databases. For *CFTR* we have used CFTR2 (<https://cftr2.org/>); for *HBB*, HbVar (<http://globin.cse.psu.edu/hbvar/menu.html>); for *GJB2*, Deafness Variation Database (<http://deafnessvariationdatabase.org/>); for *MEFV*, Infevers (<https://infevers.umi-montpellier.fr/web/>); and for *HFE*, LOVD-HFE (<https://databases.lovd.nl/shared/genes/HFE>). The same variants were searched in gnomAD v3 and counts per population were used to calculate a frequency per population (number of genotypes fixed at 71700). Incidence was calculated without correction for penetrance, assuming panmixia and even distribution between sexes. We combined counts of heterozygotes (independently for each variant within a locus, as observed) and number of individuals. The fraction of carriers within each sample was squared (providing a fraction of possible couples of carriers) and divided by four (an offspring of 25% of compound heterozygotes or homozygotes).”**

In the Methods Section of the Main Text:

“**Incidence of selected recessive disorders was calculated using direct count of pathogenic alleles after manual curation and verification of pathogenicity in Locus Specific Databases (Supplementary Information 4.6). We have calculated the expected incidence based on mode of inheritance pattern and assuming panmixia, all within cohorts (SABE versus gnomAD). Summary of steps and workflows can be found in Supplementary Information and Supplementary Fig. 4-5.**”

I would like the authors to include the following suggestions in a revised version.

Line 97-99: How unaffected elderly individuals can help to improve diagnosis accuracy of rare diseases? In my opinion, the NGS can help to improve diagnosis and the use of large datasets of unaffected individuals helps to estimate the prevalence of these diseases in specific populations.

R12) As most rare genetic disorders are early and adult-onset, the age at onset versus age of unaffected individuals for control is critical in providing an accurate diagnosis. Most datasets are composed by adults, and depending on the phenotype of interest, some individuals that carry pathogenic variants might manifest the condition after inclusion on the dataset. Even in age-related disorders (such as neurodegenerative diseases) and also in common disorders with higher prevalence among elderly (such as hypertension), the unaffected status may provide

additional power to case-control studies (Naslavsky et al., 2017; Pinese et al., 2020) in a similar design of extreme selection of phenotypes (Peloso et al., 2016 EJHG). To clarify this reasoning we added the sentences below (in red) in the Main text.

Action to R12) Lines 97-99 now read:

“Moreover, the penetrance of variants may vary across backgrounds^{14,15}. For variants associated with monogenic early and **adult**-onset disorders, unaffected elderly individuals serve as a proper control group to improve diagnosis accuracy. **Since many diseases manifest later in life, datasets composed by adults can include carriers that may express some or full clinical phenotypes. Even studies on late-onset diseases can be powered by a control group of verified unaffected status when aged older than average age at onset.**”

Line 104: Sentence “immigrant descendants from different continents and individuals from various Brazilian states” has to be clarified. It is unclear if all participants are Latin Americans or some of them are from different continents living in Brazil.

R13) SABE is a census-based cohort, therefore it is composed of residents of São Paulo city withdrawn from the 2000, 2005 and 2010 census (depending on their year of inclusion). There are 66 individuals born abroad (55 individuals from Europe and 11 from Japan). We changed the sentence to clarify the information (below, in red).

Action to R13) Line 104 now reads:

“Here we present the first high-coverage WGS of a Latin American census-based cohort composed of 1,171 unrelated elderly from São Paulo, Brazil’s largest metropolis. **Among the residents, there are** immigrant descendants from different continents, individuals from various Brazilian states¹⁸, **and 66 individuals born abroad, mostly in Europe and Japan.**”

Line 106: Please cite a reference for SABE study.

R14) We included the reference Lebrão et al (2019) in line 106.

Line 122: Please add a brief description of the pipeline used to analyze the 1171 genomes, a measure of dispersion should be added to the average data.

R15) We changed the sentence to add a measure of dispersion and a reference of how the data was generated (below in red). Complete whole-genome sequencing and bioinformatics pipeline, including quality control, are described in the methods and supplementary information (sections 1 and 2).

Action to R15: Line 122 now reads:

“High-coverage WGS data (average **38.6±6X**) was generated **using a previously described protocol**^[Telenti et al., 2016] and analyzed (Supplementary Fig. 2, Supplementary Table 3).”

Line 125: According to section methods the authors annotate the variants with Annovar, the authors could be more explicit about the number and the consequences of the variants identified in this study (missense/silent, LOF, frameshift, etc).

R16) Consequences and respective counts were described in Supplementary Table 4. Since it is a large table, we have decided it would be best fit in the Supplementary Information file.

Line 129: Change global ancestries by genetic structure. In the corresponding section the authors should discuss how individuals with ancestry mainly European and African (and with a low proportion of Native American ancestry) can represent the Latin American population.

R17) In this study, we have estimated global ancestry as one component of the genetic structure of a given population. In the referred sentence we are specifically presenting the overall proportions of ancestries as inferred by the ADMIXTURE program that are consistent with historic evidence and previous studies in population genomics of Brazilian individuals. Further insights on population structure and comparison with other Latin American populations will be explored in another manuscript.

Line 129-134: Could the authors add a description of the method used to determine the genetic structure of your cohort. Are relevant the PCA plot presented in Fig 1B? Also, a description of how was calculated is needed.

R18) We have used ADMIXTURE program to estimate individual level global ancestries as described in the Methods section of the main manuscript, where reference datasets were also disclosed. We also performed PCA with R package SNPRelate to graphically place SABE individuals among parental populations in order to demonstrate the heterogeneity and the high level of admixture, and, to some extent capture dispersion among individuals that might not be captured by the summarized proportions of ADMIXTURE analysis. Overall, however, ADMIXTURE and PCA analyses can be interpreted as redundant results. As described in the Methods section, both analyses were performed with a subset of high-confidence SNPs obtained after LD pruning. We added more details about the methods (below in red) in the Methods section.

Action to R17

We used ADMIXTURE v.1.3.0⁹ to perform global ancestry inference through supervised analysis ($K = 4$) **and 2000 bootstrap replicates, which uses a maximum likelihood framework, based on multilocus SNP genotypes.** African (N=504), European (N=503), and East Asian (N=400) **non-admixed** samples from 1KGP3, and Native Americans (N=221) from recently published datasets¹⁰, were used as parental populations (Supplementary Table 5). The Native American samples were genotyped on the Illumina Omni 2.5M array; thus the genetic variants of the 1KGP3 and SABE samples (dataset of PASS (GATK) and vSR (CEGH Filter, Supplementary Fig. 3) variants with genotypes flagged by CEGH-Filter as FD or FB set as missing) were filtered to overlap with this array, totaling 1,842,125 SNPs. LD-pruning on this subset of markers was performed with PLINK v.1.9¹¹, with an r^2 threshold of 0.1 within a sliding window of 50Kb and a shift step of 10Kb, resulting in 372,527 SNPs. We **also** used the same **LD-pruned** dataset to perform PCA analysis with **R package** SNPRelate¹².

Line 134-137: This sentence is no clear.

R19) We changed the sentence to (in red):

The **average** proportions of ancestry significantly **vary among** self-reported ethn racial groups (one way ANOVA p-value <0.0001; Extended Data Fig. 2). **Yet, 37% of the variation in the ancestry among individuals is not explained by self-reported categories** ($r^2=0.63$; p-value < 2.2e-16; fit linear model d.f. 1083). **Thus, although there is a correlation between ancestry and self-declaration categories, they are not able to capture accurate information about the heterogeneity and proportions of individuals' genetic ancestry.**

Figure 1: Please include the parental populations used as panel for the Admixture K= 4. Why the sample size for SABE cohort changed? Please justify. Except for Asian ancestry, it is difficult to know what ancestry are representing the colors. In general, for panels A & B, I suggest to use contrasting colors. Authors should be particularly careful using those labels in Fig1A.

R20) Thank you for the suggestions. We detailed the parental populations used in the Methods section and Supplementary Table 5. We added this information on line 130 (R20a below, in red). Thank you observation on the sample size change. We performed the global ancestry analyses for all individuals, from this total, there were three individuals self-reported as Indigenous, and we opted to remove them from the figure because it was very difficult to display them in it. We have included the justification in the Figure legend (R20b below, in red). Figure labels serve for 1A and 1B, but we have made them larger for clarification. We agree that choice of colors is important. Since there are no overlaps in the Native American parental group observable on PC1 and PC2 components (1B), we opted to maintain the colors since there is a clear background of Native American components in most non-Asian individuals (averaging nearly 7%).

Actions to R20:

R20a) Line 130 now read:

The average global ancestries for SABE are 0.726 ± 0.263 European, 0.178 ± 0.209 African, 0.067 ± 0.066 Native American, and 0.028 ± 0.162 East Asian (Fig.1A, **Methods**, Supplementary Table 5).

R20b) Figure 1 legend now reads:

Figure 1. Global ancestry of SABE cohort. A. Individual ancestry bar plots of SABE cohort (N = 1,168) using Europeans (EUR), Africans (AFR), East Asians (EAS), and Native Americans (NAM) as parental populations and distributed by self-reported ethn racial groups (Supplementary Figure 5). NA = Not available. B Principal component analysis of SABE individuals and parental populations. **Analyses were performed with 372,527 SNVs (after overlapping- and LD-pruning). Three individuals self-reported as Indigenous had a high degree of admixture but were removed from Figure 1 due to the small sample size of the group. AFR, EAS and EUR from 1KGP3 and NAM from Borda et al., (2020). Specific samples are described in Supplementary Table 5.**

Line 164: I recognize the hard-manual curation work, so a diagram or more details about this process should be added. It would also be relevant to compare these results with those derived from automated annotators such as InterVar (10.1016/j.ajhg.2017.01.004).

R21) We have added details of the manual curation in the Supplementary Information as explained in R5. Although we agree that for most ACMG criteria, InterVar, VarSome or other automated pathogenicity annotators are very useful and were applied to our data, specific criteria such as literature evidence of co-segregation and functional assays (still non standardized) or details on submission are needed for a complete classification. In summary, even with all possible ACMG criteria fulfilled, the automated annotators are still not providing the full evaluation. Efforts in literature data mining are being attempted, VarSome being one example, but so far there is no automated annotator, to the best of our knowledge, that can assess criteria such as segregation in described families (PP1), frequency in affected cohorts (PS4), or functional studies not restricted to ClinVar reports (PS3).

Line 174-178: The authors claim about incidence estimation for autosomal recessive diseases using population-based genetic data, nevertheless it is hard to know how these values were obtained due the lack of a description of the method used to obtain the incidence, are you following the pipeline reported by Bainbridge M? (<https://doi.org/10.1007/s00439-020-02135-5>). I encourage the authors to include a method section describing the methodological procedures for these estimations and to show the values of the incidence estimated in the main text.

R22) Thank you for your suggestions. We have included a detailed description on the incidence estimation within the Supplementary Information section (see R13). The incidence was calculated using direct count of pathogenic alleles after verification of pathogenicity in Locus Specific Databases. We have calculated the expected incidence based on mode of inheritance pattern and assuming panmixia, all within cohorts (SABE versus gnomAD). It is noteworthy to mention that we have only included known pathogenic mutations (from the ‘head’ alleles up to the rare alleles on the ‘tail’), as also assumed by Bainbridge, 2020 (Human Genetics), whose main goal is to propose a framework where MAF and counts of known pathogenic alleles are used upon availability in datasets.

We have included in Main text the underestimation comment due to unobserved pathogenic alleles driven by classification challenges, as stressed by Bainbridge (below, in red).

Action to R22:

R22a) Lines 182-185 now reads:

“These disparities observed for GJB2 and MEFV between Brazilians and global gnomAD, but similar to gnomAD Latinos and PAGE Study samples of Cubans, Puerto Ricans, and Central Americans are probably due to the Iberian, Mediterranean, and Middle Eastern contributions in Brazil, **but we cannot exclude that penetrance of such variants may be lower than previously estimated.**

Estimating incidence of recessive disorders is challenging due to ascertainment of unrelated individuals within a given population-based sample and classification of pathogenicity, since most pathogenic variants are rare and the distribution of variants

within a population is not known a priori [Bainbridge, 2020]. Our results are limited to observed pathogenic alleles (usually the more frequent and a few rare ones) curated by locus-specific databases to provide a comparison of expected individuals in homozygous and biallelic states for selected recessive disorders.”

R22b: The following sentence was added to Methods:

Incidence of selected recessive disorders was calculated using direct count of pathogenic alleles after manual curation and verification of pathogenicity in Locus Specific Databases (Supplementary Information 4.6). We have calculated the expected incidence based on mode of inheritance pattern and assuming panmixia, all within cohorts (SABE versus gnomAD). Summary of steps and workflows can be found in **Supplementary Information and Supplementary Fig. 4-5.**

Would it not have been useful to have family-based data to track for the transmission of low frequency causal variants in pedigrees to see if any segregate with disease?

R23) Yes, we agree. Until now, the SABE Study has not collected data from other family members. To provide an in-depth analysis of segregation of pathogenic alleles and corresponding phenotypes, we have submitted a family-based follow up to the IRB.

Line 224: Change indicating by suggesting.

R24) Change applied.

Line 263-264: Please add a brief description about the method used to perform de novo assembly.

R25) Methods section of main text contains the full pipeline. We have included the following sentence to the Non-Reference Sequences (NRS) in Main text (below, in red).

Action to R25: Lines 263-264 now reads:

“We characterized these ‘missing’ segments by performing de novo assembly of high-quality reads that do not map to current reference **using a pipeline of assemblers, aligners and mappers, with parameters set to reduce false positives (Methods).**”

Figure 4.- Please specify the letters for each panel

R26) Thank you, we have added the letters for each panel in Figure 4.

Figure 5.- The same comment as figure 4

R27) Letters were small and not standard with the other figures. We have added the letters in standard.

Line 369-380. The GWAS story is hard to understand. In general, this section need to be improved. They performed GWAS for selected phenotype (BMI, LDL, triglycerides, positive history of cancer, cognitive decline, diabetes, frailty, and hypertension). Could be more explicit on the phenotypes used in GWAS? What about the possibility of undiagnosed cases among controls? Please mention the cutoffs to define the phenotypes. There is no mention of the statistical power. If the characteristics of the 1171 individuals were stratified by case or control, and then by gender and for multiple comparisons, is there an adequate statistical power? Authors should add a table or regional plots for the new 12 hits. Also, expand the discussion about the possible mechanism or biological processes in which they participate. I suggest to include an enrichment pathway analyses for the new and previously described hits. Finally, to address the association of rare variants with the studied phenotypes I suggests a sequence Kernel analysis.

R28) Thank you for your comment and suggestions. As mentioned in R1, we have decided to remove the GWAS section altogether from this manuscript. To provide appropriate validation and discussion of findings, a future manuscript is planned to include GWAS and burden/rare variant collapsing analyses, including downstream analyses such as replication and functional annotation of findings.

Line 382-406. In general, the discussion does not add much to the previously discussed in other sections.

R29) As mentioned above, this is a descriptive manuscript. Each section contains links to corresponding literature on the field.

Line 519-758. In the methods section please add a section about the estimation of incidence of rare diseases.

R30) We have added the incidence estimation on Methods and details in the Supplementary Information file (Response 13)

Line 565. Based on what criteria the authors chose the reference population panel for admixture. The authors have to be cautious about supervised analysis due it is only suitable when the reference individuals can be assigned to ancestral populations with certainty and ancestral populations are fairly homogeneous. In this case, several populations from the 1KGP3, shows gene flow between them, causing that the ancestry fraction estimates in target dataset can suffer from bias.

R31) Reference populations were chosen upon availability and previous studies on population genomics of Brazilians.

We agree with the reviewer, the choice of parental populations is fundamental to the accuracy of the analyses. Most studies with a Brazilian population use the tri-hybrid model of parental population (AFR, EUR and NAM) [Kehdy et al., 2015]. Here we chose to add EAS, due to the high proportion of Asian immigrants (especially from Japan) coming to São Paulo. São Paulo also received immigrants from the Middle East [Centro de Documentação e Disseminação de Informações (Brazil). *Brazil, 500 years of settlement*, IBGE, 2007], but the method applied does not distinguish accurately Middle Eastern from European ancestries, since we have not included Middle Eastern samples as a parental population.

Before running ADMIXTURE in the SABE sample, we performed unsupervised runs on ADMIXTURE using only parental populations to check if they formed distinct clusters and if there were any admixed individuals between them. In this way, we can use these contemporary individuals as a proxy for ancestral populations.

Action to R31:

In the Methods section, we have added the following sentences

We used ADMIXTURE v.1.3.0⁹ to perform global ancestry inference through supervised analysis ($K = 4$) **and 2000 bootstrap replicates, which uses a maximum likelihood framework, based on multilocus SNP genotypes.** African (AFR, $N=504$), European (EUR, $N=503$), and East Asian (EAS, $N=400$) **non-admixed** samples from 1KGP3, and Native Americans (NAM, $N=221$) from recently published datasets¹⁰, were used as parental populations (Supplementary Table 5). The Native American samples were genotyped on the Illumina Omni 2.5M array; thus the genetic variants of the 1KGP3 and SABE samples (dataset of PASS (GATK) and vSR (CEGH Filter, Supplementary Fig. 3) variants with genotypes flagged by CEGH-Filter as FD or FB set as missing) were filtered to overlap with this array, totaling 1,842,125 SNPs. LD-pruning on this subset of markers was performed with PLINK v.1.9¹¹, with an r^2 threshold of 0.1 within a sliding window of 50Kb and a shift step of 10Kb, resulting in 372,527 SNPs. We **also** used the same **LD-pruned** dataset to perform PCA analysis with **R package** SNPRelate¹². **Our choice of parental populations to maximize ancestry inference accuracy was based on the tri-hybrid model (AFR, EUR and NAM) previously used by other authors [Kehdy et al., 2015], which converges with historical evidence. We added EAS due to the high proportion of Asian immigrants (mainly from Japan) settled in São Paulo [Centro de Documentação e Disseminação de Informações (Brazil). Brazil, 500 years of settlement, IBGE, 2007]. Before running ADMIXTURE in the SABE sample, we performed unsupervised analysis (K=4) using only parental populations to check if they formed distinct clusters and if there were admixed individuals between them. In this way, we can use these contemporary individuals as a proxy for ancestral populations.**

Line576. Did the authors use the PCAs calculated with the reference panel for ancestry correction in GWAS analyses? If true, you need to perform a new PCA estimations without the reference panel and use it to perform the ancestry adjustment.

R32) Even though we have used PCs derived from sample-only PCA, further adjustments will be applied in a different manuscript. As mentioned in R1, we have removed the GWAS section altogether from this manuscript, including line 576.

Line 680. Change SNP's by SNVs. In general, change SNP by SNV in the full text.

R33) We have replaced SNP by SNV in the Main manuscript and in the Supplementary Information.

Reviewer #3 (Remarks to the Author):

In their manuscript, Naslavsky et al. present high-coverage WGS data for over 1100 elderly, Brazilian individuals. With these data, they identify ~2M genetic variants absent from other large public databases, including novel mobile elements, nonreference sequences, and over 140 novel alleles from HLA genes. They also reassess pathogenicity of variants associated with Mendelian disorders, calculate the expected incidence for selected recessive disorders in the Brazilian population, provide an improved panel for whole-genome and HLA imputation, and perform GWAS for ~10 traits. These findings are of potential interests for those in the fields of human genetics, genomics, population genetics, and anthropology, among others. I found all statistical methods to be valid and appropriate.

In my opinion, the strength of their work lies on the generation of a dataset comprising high-coverage WGS for over 1100 individuals from an underrepresented population in genomics, with associated data on medically-relevant phenotypes. As the authors note, currently available public genomic datasets lack non-European and admixed populations and this lack of diversity may lead to biases in precision medicine and consequently less accurate genetic tests on underrepresented populations, potentially increasing health disparities and affecting historically marginalized communities. I believe this study is very timely and its dataset is an interesting resource for research in human genetics and genomics.

Please find additional comments below. Thank you!

Best regards,

C. Eduardo Guerra Amorim

Minor comments:

a. Lines 95-96: Perhaps it would be good to cite some references from the literature that also discuss this relationship between expected and observed frequencies, disease prevalence, etc.

R34) Thank you for your suggestion. We have cited three references along the results/discussion section about this rationale. First, the recommendations of ACMG for benign evidence when population frequency of candidate variant is incompatible with the incidence (Richards et al, 2015), which also promotes stand-alone benign assertion when MAF reaches given threshold (discussed in recent publications with population-specific variants classified as P/LP but with higher frequencies. Shah and colleagues (2018) use frequency as a baseline for variant reclassification, as did Xiang and colleagues (2020), who systematically downgraded due to frequency. We included the citation of these references in the Introduction (below, in red).

Action to R34:

Lines 95-96 now reads:

“Knowledge about allelic frequencies from multiple populations is also crucial when prioritizing candidate clinical variants. For rare Mendelian disorders, the frequency **of a pathogenic variant in any given population cannot be higher than the incidence of its associated disease, considering compatibility with mode of inheritance and penetrance**[Richards et al, 2015; Shah et al, 2018; Xiang et al, 2020].”

b. Data availability: It reads on lines 113-114 that variants and allele frequencies are publicly available. I wonder whether this journal requires the full dataset (fastq, VCF or BAM files) to be available.

R35) Although the journal accepts publications on private datasets, we agree that sharing individual level data promotes collaboration and full usage of data, as well as foster reproducibility of findings. Due to IRB constraints, many datasets are to remain private. We are submitting the sequencing datasets (BAM files) and variant calling datasets (gVCF files) to EGA in controlled access mode (addressed on R1).

c. Lines 134-137: What does partially account for ancestry variation? Is that the self-reported ethnoracial group? Can you briefly explain the model? Preferably in the main text or perhaps in Extended Data Figure 2 title? (Please check for typos e.g. “withing”)

R36) Thank you for pointing to the typo in the legend of Extended Data Fig. 2, we have corrected it. Due to the high degree of admixture, self-reported ethnoracial groups are not fully correlated to a single ancestry or composition of ancestries. There is however a significant difference among groups regarding European and African ancestry proportions. Aligned with previous responses R19 and R23, we have addressed that in Extended Data Fig. 2 and added the following sentences to the Main text:

The **average** proportions of ancestry significantly **vary among** self-reported ethnoracial groups (one way ANOVA p-value <0.0001; Extended Data Fig. 2). **Yet, 37% of the variation in the ancestry among individuals is not explained by self-reported categories** ($r^2=0.63$; p-value < 2.2e-16; fit linear model d.f. 1083). **Thus, although there is a correlation between ancestry and self-declaration categories, they are not able to capture accurate information about the heterogeneity and proportions of individuals' genetic ancestry.**

d. Line 159: missing word after “curated” (?)

R37) Thank you for observing. We have addressed this paragraph in response R5a. Now it reads:

“We analyzed ‘Pathogenic’ or ‘Likely Pathogenic’ (P/LP) ClinVar asserted variants carried by SAGE individuals across 4,250 genes associated with monogenic disorders (Online Mendelian Inheritance in Man - OMIM disease genes, Supplementary Table 6) and manually curated **the variants in genes associated with dominant inheritance** using ACMG guidelines¹⁴ **and in-depth literature support, performed by clinical geneticists (complete workflow on filtering, annotation and counts are described in** Supplementary Fig. 4 **and Supplementary Information 4).**”

e. Lines 162-163: Do you mean you could only confirm the effect of 3% of the studied mutations? This number is surprisingly small! This seems to be an interesting result. Have previous studies also examined that for Mendelian diseases? Please add some references in the main text to help readers assess the importance of these findings.

R38) These 3% variants are carried by individuals that reported compatible phenotypes. There is a chance that among variants classified as having an incomplete penetrance and/or subclinical mild phenotypes there are individuals who indeed manifest endophenotypes not available in the dataset or not yet observed by them. This is indeed an expected finding already described in the literature: from counting knockouts in healthy individuals as shown by MacArthur et al, 2012, estimating resilient individuals within large datasets (Chen et al, 2016) and using pLOF prevalences to model gene intolerance to deleterious mutations (Lek et al, 2016; Karczewski et al, 2020). However, assessing elderly individuals from a census-based cohort may provide the ideal setup to investigate this topic. The rationale was previously applied by our group with a subset of SABE participants submitted to whole-exome sequencing (Naslavsky et al, 2017) and recently by whole-genome sequencing Australian elderly (Pinese et al, 2020). Indeed, among ACMG 59 actionable genes, we have found a comparable proportion of carriers in SABE elderly (1.2%) versus Australian elderly (1.1%) (Pinese et al, 2020).

We have added the following sentences to the Main text:

It has been reported that large datasets contain pathogenic mutations that can be harbored by unaffected individuals, as shown by Chen and colleagues after deeply screening genes associated with monogenic early-onset disorders[Chen et al, 2016]. Healthy elderly individuals from Australia are reportedly depleted of disease-causing variants, but still carry clinically relevant mutations[Pinese et al, 2020]. It is noteworthy that pathogenicity misclassification itself can alter estimations of disease prevalence[Shah et al, 2018]. Manual curation promotes the downgrading of pathogenic assertions when diverse ancestries are added to databases.

f. Paragraph starting in line 174: How do you calculate the expected incidence? Is that simply the number of homozygotes and compound heterozygotes? Or the number of expected offspring? Or just that the allele frequencies are similar between datasets?

R39) Yes, it is indeed a direct count procedure assuming random mating of carriers, after curation of pathogenic variants. We have addressed the method in R13, followed by additional paragraphs in Supplementary Information.

g. Lines 196-199: Are all these polymorphic in humans?

R40)

We thank the reviewer for this question. Not all MEIs (7490 in total) are polymorphic in humans. We expected that at least 5,571 (74.3%) events (classified as "Shared" by us) should be polymorphic in humans. To remind the reviewer: "Shared" are those MEIs present in two or more unrelated SABE individuals and also in individuals from gnomAD). To make it clear, we added a comment in the Methods text regarding this point:

"SABE events found in DGV were classified as Shared MEIs **and are potentially polymorphic in humans**".

h. Figure 5: Correct me if I am wrong, but I believe “America” should be replaced for “South America” or “Americas.”

R41) Thank you for observing. We have changed that to "Americas".

i. Can you give some reasoning for why relatively common variants (allele frequency >2%) are more often missing from other datasets than less common variants (AF 0.5% - 2%), but not very rare ones (AF <0.5%)?

R42) The relative proportions demonstrate a general (expected) trend of common variants being previously described whereas rare variants have a higher proportion of newly described variants. Figure 5B X-axis has frequency bins that are not proportional to the number of individuals, in addition to the different number of individuals in SABE (n=1171) vs. 1KGP3 (n=2504) which lead to fixed bins (0.5-1% or 1-2%) having small number of variants. We wanted to emphasize the switch in relative proportions of singleton to 0.5% versus >2% HLA variants.

Actions to R42:

Addition of X-axis label to Figure 5B, bigger letters to each panel and legend changes.

j. Line 568 (Methods): reference 10 seems to be focused on African ancestry. Please check if this reference is correct.

R43) Thank you for noticing, we have indicated a wrong reference. It should be Borda et al. 2020 published in bioRxiv. We updated this reference for the peer-reviewed version now published in PNAS. Now it reads:

We used ADMIXTURE v.1.3.0⁹ to perform global ancestry inference through supervised analysis (K = 4). African (N=504), European (N=503), and East Asian (N=400) samples from 1KGP3, and Native Americans (N=221) from recently published datasets [**Borda et al, 2020**], were used as parental populations (Supplementary Table 5).

k. Last, I think the paper is generally well written, but some sentences are wordy and confusing. Below I show some examples:

(1) Lines 88-89: “Most importantly ... harbor specific variants.”

R44) We have removed “Most importantly”

(2) Lines 95-96: The frequency of what? The observed allele frequency? Do you mean the observed allele frequency cannot be larger than the expected frequency calculated based on disease incidence? Is this true for both recessive and dominant diseases? Would incomplete penetrance, complementation, advantage of the heterozygote (in recessive disorders), and late disease onset affect this relationship between expected and observed frequencies?

R45) Thank you for observing. We have addressed that in R38 adding to the initial sentence:

Knowledge about allelic frequencies from multiple populations is also crucial when prioritizing candidate clinical variants. For rare Mendelian disorders, the frequency **of a pathogenic variant in any given population cannot be higher than the incidence of its associated disease, considering compatibility with mode of inheritance and penetrance [Richards et al, 2015; Shah et al, 2018; Xiang et al, 2020].**

(3) Lines 159-163: What do you mean with curation has led to reclassification of pathogenicity *by* inheritance mechanism or penetrance?

R46) There were three situations regarding inheritance mechanisms: a large number of genes annotated as having both dominant and recessive modes of inheritance were (1) observed to have a single mode (recessive); (2) lacking information on the second pathogenic allele in trans (but with orthogonal evidence of haploinsufficiency); or (3) conditioned pathogenicity when the allele in trans is a pLOF (hereby classified as ‘recessive allele’).

As addressed in R5b, Supplementary Information 4.4. now read:

“In order to identify individuals carrying variants with potential clinical implications, including the reassessment of related phenotypes to support the analyses, we have filtered a total of 394 variants asserted as either ‘Pathogenic’ or ‘Likely Pathogenic’ (P/LP) in genes annotated to have a dominant mode of inheritance only, and in genes with more than one mode of inheritance, including dominant or monoallelic. Manual curation aiming reclassification of pathogenicity using ACMG criteria was performed **by two independent clinical geneticists (professionals with MD and previous experience in variant pathogenicity classification using ACMG criteria and patient clinical information). Manual curation included family-based information described in the available literature, evidence details on the original assertions, and allele frequency. Each of the 394 variants in dominant genes containing P/LP ClinVar assertions was submitted to manual curation aided by population-specific frequencies (gnomAD and SABE, mainly); ClinGen (to reannotate inheritance modes); review of VarSome automated calculation of ACMG classification criteria; and in-depth analyses on ClinVar submissions leading to classification, particularly in evidence levels (ACMG criteria assigned and provided by submitters, to adjust PP5), details on co-segregation of ClinVar assertion combined with literature reports of carriers and families (to adjust PP1). OMIM aided reclassification of gene’s mode of inheritance in cases where only one affected case was reported and recessive mode could not be excluded, or else when pLOF consequence would only be detected in trans with another P/LP variant (hereby classified as ‘recessive allele’, indicating haploinsufficiency).** A total of **116** variants (**29%**) were reclassified as non-pathogenic assertions (benign, likely benign or unknown significance) (**Supplementary Table 11**), **most of which had no assertion criteria provided (70 variants), 44 had criteria provided by a single submitter and 9 by multiple submitters.** The remaining **278** kept as pathogenic or likely pathogenic (**Supplementary Table 12**). Among the latter, literature validation and matching

phenotypes, when available, enabled further characterization of variants to either a reported reduced penetrance, non-dominant mode (of the specific allele or gene), or associated to clinical features that are not severe enough to cause mortality before the average age of subjects (Extended Data Tab. 1).”

Supplementary Table 11. Reclassified variants after manual curation in genes with dominant mode of inheritance.

Large table displayed only on Supplementary Tables worksheet.

Supplementary Table 12. Pathogenic variants and categories after manual curation in genes with dominant mode of inheritance.

Large table displayed only on Supplementary Tables worksheet.

(4) Lines 164-165: “Manual curation promotes ... to databases.”

R47) We have rewritten to:

“Manual curation promotes the downgrading of pathogenic assertions when diverse ancestries are added to databases **and variants of higher frequency are identified.**”

(5) Lines 171-173: “Also, regarding P/LP ... elderly cohort.”

R48) We have rewritten to:

“Also, regarding P/LP asserted variants in the 59 ACMG actionable genes list (Supplementary Table 11), 14 were found in 1.2% of individuals²⁶ comparable to the Australian elderly cohort¹⁷, **demonstrating that clinically relevant variants are detectable at low but equivalent proportions.**”

REVIEWER COMMENTS

Reviewer #1 (Remarks to the Author):

Thank you for your comprehensive and thoughtful responses to my comments. All comments have been addressed and I have no further issues to raise.

Sincerely,

Mark Pinese

Reviewer #2 (Remarks to the Author):

In the revised version of the paper “Whole-genome sequencing of 1,171 elderly admixed individuals from the largest Latin American metropolis (São Paulo, Brazil)”, the authors improved the quality of the manuscript adding a better description of the methods and a better description of the results (for sections that they decided to maintain in the manuscript). Nevertheless, I have still additional comments and minor remarks.

I would like the authors add the GWAS section again, with the improvements previously suggested because the current results are so similar to the paper Naslavsky et al. Hum Mutat 2017. In my opinion, the exclusion of the GWAS results decreases the quality and novelty of the article.

Due to the relevance of the reported 2 million of new variants and the new structure of the article, a similar figure to Figure 2 should be made to highlight this finding.

Line 97. Please exchange associateddisease by associated disease.

Line 126-127. Please improve the redaction of the sentence.

Line 138-145. Although the authors added a better description of the results, there is still unclear how they made the correlation between self-reported ancestry and estimated ancestry. How did the authors assign the values to each individual's self-reported ancestry?

Figure 1. Panel A. I have the same concerns as in my previous revision. Please reconsider to add to the plot the parental populations to help the readers to improve the plot understanding. Panel B. Again, is this plot relevant? If the answer is yes, there is no mention or discussion about this result in the main text.

Line 194-198. Please add the values of the estimated incidences of rare diseases in the main text.

Line 276-278. Provide a reference for this sentence.

Line 279-278. The authors should be more explicit about the pipeline, assemblers and other bioinformatic tools used for the de novo genome assembly.

Line 331-33. The authors claim that “The SABE+1KGP3 reference panel improved imputation independent of the target cohort and its level of admixture, suggesting that our panel can improve imputation for other Latin American populations”. The authors can demonstrate this using other published datasets of Latin-Americans populations with a different demographic history, such like Mexicans or Peruvians.

Figure 4. Improve the figure 4

Figure 5. Please improve the figure, it is difficult to find the panel’s letters.

Discussion. In general, the DISCUSSION is repetitive and does not add much to what was previously discussed in the results.

Reviewer #3 (Remarks to the Author):

All of my concerns have been addressed in this revised version of the manuscript. Overall, I consider the manuscript much improved after revisions.

Responses to Reviewer comments

We have addressed the Reviewer's comments below. Each response is numbered and whenever sentences are added or altered in the Main text, Methods or Supplementary Information, we have (1) tracked changes to the submitted files (except changes in table renumberings, which were applied), (2) copied the context of the alterations and pointed the lines of the original manuscript to this file, and (3) highlighted the editions in bold/red font.

In the review process we noticed that the imputation histogram was drawn for the total number of imputed variants and not for the total number of variants with info score ≥ 0.8 across the allele frequency spectrum (Figure 4B, and Figures S9B to S23B). This error did not change any result or interpretation presented in the text. We replaced with the correct figures accordingly in the reviewed files.

REVIEWER COMMENTS

Reviewer #1 (Remarks to the Author):

Thank you for your comprehensive and thoughtful responses to my comments. All comments have been addressed and I have no further issues to raise.

Sincerely,

Mark Pinese

Reviewer #3 (Remarks to the Author):

All of my concerns have been addressed in this revised version of the manuscript. Overall, I consider the manuscript much improved after revisions.

R1) We thank reviewers 1 and 3 and invite them to check the latest revisions implemented to this submission. Most are minor corrections in text format. As a highlight, we have performed an additional imputation experiment targeting Guatemalan and Peruvian samples to evaluate the addition of SABE1171 to 1KGP3 reference panel (Supplementary Tables 22-23, and Supplementary figures 24-31).

Reviewer #2 (Remarks to the Author):

In the revised version of the paper "Whole-genome sequencing of 1,171 elderly admixed individuals from the largest Latin American metropolis (São Paulo, Brazil)", the authors improved the quality of the manuscript adding a better description of the methods and a better description of the results (for sections that they decided to maintain in the manuscript). Nevertheless, I have still additional comments and minor remarks.

I would like the authors add the GWAS section again, with the improvements previously suggested because the current results are so similar to the paper Naslavsky et al. Hum Mutat 2017. In my opinion, the exclusion of the GWAS results decreases the quality and novelty of the article.

Due to the relevance of the reported 2 million of new variants and the new structure of the article, a similar figure to Figure 2 should be made to highlight this finding.

R2) We thank Reviewer 2 for the comments and suggestions. Except for the GWAS section, all comments and suggestions were addressed as seen below, including the addition of non-Brazilian Latin American samples as imputation targets (see R10).

Authors agreed with the editorial suggestion to maintain the GWAS section removed since validation steps would take longer due to access to other datasets. In particular, authors agreed that digging further from the comparison of common-variant GWAS versus WGS-based GWAS to conducting the full expected pipeline of association studies would deviate from the main message of this manuscript, with the possibility of insufficient power to detect signals due to relatively small sample size.

Following your suggestion, we have included in Extended Figure 1A a diagram containing summarized counts of total and 'novel' variants in a similar fashion as Figure 2 from Naslavsky et al. Hum Mutat 2017.

Line 97. Please exchange associateddisease by associated disease.

R3) Correction applied (line 99).

Line 126-127. Please improve the redaction of the sentence.

R4) We have rewritten the sentence and expanded the description of SABE cohort collections.

Line 127-131 now reads

Data collection [Lebrão et al, 2019] involves at-home interviews with 11-section questionnaires, including cognitive screening, self-reported race/ethnicity status and standard tests of over 20 health conditions, habits and phenotypes, medication inventory, and functional measurements, such as frailty, dexterity, balance and mobility summarized in Supplementary Table 2.

Line 138-145. Although the authors added a better description of the results, there is still unclear how they made the correlation between self-reported ancestry and estimated ancestry. How did the authors assign the values to each individual's self-reported ancestry?

R5) We have added the sentences below (lines 144-155).

As part of the interview process, all individuals were asked to self-report to one of the ethnoracial groups routinely used by the Brazilian Institute of Geography and Statistics in the national census [Reference link: https://censo2010.ibge.gov.br/images/pdf/censo2010/questionarios/questionario_basico_cd2010.pdf] (White, Black, Yellow, Pardo – translated as Mixed, or Indigenous). The average proportions of genetic ancestry significantly vary among self-reported ethnoracial groups (one way ANOVA p-value <0.0001; Extended Data Fig. 2A). Yet, 37% of the variation in

European ancestry among individuals is not explained by self-reported **ethnoracial groups** ($r^2=0.63$; p -value $< 2.2e-16$; fit linear model d.f. 1083). Thus, although there is a correlation between **genetic** ancestry and self-reported **ethnoracial groups**, they are not able to capture accurate information about the heterogeneity and proportions of individuals' genetic ancestry. In addition, three individuals self-reported as Indigenous **had a high** degree of admixture but were removed from Figure 1 due to the small sample size of the group.

Figure 1. Panel A. I have the same concerns as in my previous revision. Please reconsider to add to the plot the parental populations to help the readers to improve the plot understanding. Panel B. Again, is this plot relevant? If the answer is yes, there is no mention or discussion about this result in the main text.

R6) We agree with the Reviewer that the information in figure 1B parallels that in panel 1A. Authors have decided to remove panel B from the main article structure to provide a single and direct message stated in Panel A, which was kept and to which parental population samples were added. That said, authors decided to keep it as Panel B to the Extended Data Figure 2, since it highlights the extreme variability in ancestries in a way that is familiar to many readers who use PCA approaches, conveying information beyond "average ancestry", and placement of individuals in multivariate space.

Main text, Figure 1 legend now reads:

Figure 1. Global ancestry inference of SABE cohort. Individual ancestry bar plots of SABE cohort (N = 1,168) using **supervised admixture analysis (K=4)**. Africans (AFR), Europeans (EUR), East Asians (EAS), and Native Americans (NAM) samples are used as parental populations. **SABE cohort individuals** are distributed by self-reported ethnoracial groups (**according to the Brazilian Institute of Geography and Statistics categories Asian, White, Mixed and Black; see** Supplementary Figure 5). NA = Not available.

Extended Data Figure 2 legend now reads:

Extended Data Fig. 2. Ancestry distributions and self-reported ethno-racial groups. A. Upper section: Boxplots of the proportions of genetic ancestry per self-reported ethno-racial groups (one way ANOVA p -value < 0.0001 ; Tukey test p -value < 0.001). Bottom table: Counts of individuals per self-reported ethno-racial group and corresponding average ancestries; Number of individuals within different ranges of ancestry proportions. **B. Principal component analysis of SABE individuals and parental populations. Analyses were performed with 372,527 SNVs (after overlapping- and LD-pruning). AFR, EAS and EUR from 1KGP3 and NAM from Borda et al., (2020)[ref]. Three individuals self-reported as Indigenous had a high degree of admixture but were removed due to the small sample size of the group. Specific samples are described in Supplementary Table 5.**

Main text methods now reads (line 590):

We also used the same LD-pruned dataset to perform PCA analysis with R package SNPRelate¹² (**Extended Data Fig. 2B**).

Line 194-198. Please add the values of the estimated incidences of rare diseases in the main text.

R7) We have added the values, as suggested.

In main text (lines 202-213):

Common pathogenic variants in genes associated with selected recessively inherited Mendelian disorders were manually curated using locus-specific databases and ACMG. Common and rare P/LP variants in *CFTR*, *HBB*, *GJB2*, *MEFV*, and *HFE* were accounted for incidence estimates **after calculating, from carrier frequencies, the expected offspring number of homozygotes and compound heterozygotes** (Supplementary Table 14). We showed that cystic fibrosis and hemoglobinopathies have similar expected incidences when compared to gnomAD, **respectively about one cystic fibrosis affected newborn in ten thousand births and one hemoglobinopathy affected newborn in three thousand. Estimations were calculated from *CFTR* pathogenic variant carrier frequencies of 1.8% in SABE and 2% in gnomAD (Chi-squared 0.26, 1 d.f.=1, p=0.63) and *HBB* of 3.9% in SABE and 3.4% in gnomAD (Chi-squared 0.26, 1 d.f.=1, p=0.35).** Other diseases appear more frequently in Brazilians (*GJB2*-related deafness, **one in 5.7 thousand in SABE versus one in 19 thousand in gnomAD and *MEFV* Familial Mediterranean fever, one in 55 thousand versus one in 353 thousand in gnomAD**). These disparities observed for *GJB2* and *MEFV* between Brazilians and global gnomAD, but similar to gnomAD Latinos (one in 66 thousand for *MEFV*) and PAGE Study samples of Cubans, Puerto Ricans, and Central Americans are probably due to the Iberian, Mediterranean, and Middle Eastern contributions³²⁻³⁴ in Brazil, but we cannot exclude that penetrance of such variants may be lower than previously estimated.

Line 276-278. Provide a reference for this sentence.

R8) We thank the reviewer for the useful suggestion and now included a reference for this sentence. Wong et al, 2020 (Nature Communications **11**, 5482) showed previously discarded RNAseq reads aligned to non-reference segments, presenting evidence for potential functional relevance of such findings.

Line 279-278. The authors should be more explicit about the pipeline, assemblers and other bioinformatic tools used for the de novo genome assembly.

R9) We agree with the reviewer that more details about the pipeline and tools used for the de novo assembly of NRS would be useful for the reader. We, therefore, added a schematic representation of pipeline steps (given as Supplementary Figure 8) with indication of tools used at these steps.

Line 331-33. The authors claim that “The SABE+1KGP3 reference panel improved imputation independent of the target cohort and its level of admixture, suggesting that our panel can improve imputation for other Latin American populations”. The authors can demonstrate this using other published datasets of Latin-Americans populations with a different demographic history, such like Mexicans or Peruvians.

R10) We agree with the reviewer and we evaluated imputation performance in two other admixed Latin American populations: (i) 391 Mestizos from Peru genotyped with Illumina Omni 2.5M array (Harris et al. 2018), and (ii) 640 admixed individuals from Guatemala genotyped with Infinium OncoArray-500K BeadChip (data not published). We show a general improvement of imputation using the SABE+1KGP3 reference panel compared to the 1000G reference panel, although reduced in relation to the improvement observed for Brazilian EPIGEN cohorts. As demonstrated for the EPIGEN Brazilian cohorts, we imputed more variants including high confidence imputed variants ($\text{info} \geq 0.8$) and improved imputation accuracy. The following Peruvian and Guatemalan dataset providers were added as coauthors: Carlos P. Rojas, Cesar Sanchez, Omar Caceres, and Michael Dean.

We added Supplementary Tables 22-23, Supplementary Figures 24-31, and the following wording in the Main text (lines 329-335):

We also evaluated the improvement of the SABE+1KGP3 panel independently in each EPIGEN cohort and in two other admixed Latin American populations from Peru (N=391 Mestizos[Ref: Harris et al, 2018]) and Guatemala (N=640 individuals, unpublished dataset) and also observed a general improvement of imputation (Supplementary Tables 18-23, Supplementary Fig. 9-31), although reduced for Peruvians and Guatemalans when compared to the gain observed for Brazilian EPIGEN cohorts. This improvement was also observed regardless of the chromosome tested.

Figure 4. Improve the figure 4

R11) We did not understand which type of improvement the reviewer requested. We are happy to address further with additional clarification. As mentioned, Figure 4B was altered from the previous version to correct the number of imputed variants with $\text{info} \geq 0.8$ instead of the total number of variants.

Figure 5. Please improve the figure, it is difficult to find the panel's letters.

R12) We have changed Figure 5 with larger letters indicating each panel.

Discussion. In general, the DISCUSSION is repetitive and does not add much to what was previously discussed in the results.

R13) It is noticeable that the discussion section is a short summary of previously discussed results, since we have added discussion content along the results. The authors believe that the discussion section can be kept summarized as it is.

REVIEWERS' COMMENTS

Reviewer #2 (Remarks to the Author):

Authors have adequately addressed my comments. Thanks for such an interesting paper.